# Inferring stochastic low-rank recurrent neural networks from neural data

**Matthijs Pals[1,2] A Erdem Saǧtekin[1,2,3] Felix Pei[1,2] Manuel Gloeckler[1,2] Jakob H Macke[1,2,4]**

[1]Machine Learning in Science, Excellence Cluster Machine Learning, University of Tübingen, Germany
[2]Tübingen AI Center, Tübingen, Germany
[3]Graduate Training Centre of Neuroscience, University of Tübingen, Germany
[4]Department Empirical Inference, Max Planck Institute for Intelligent Systems, Tübingen, Germany

## Abstract

A central aim in computational neuroscience is to relate the activity of large populations of neurons to an underlying dynamical system. Models of these neural dynamics should ideally be both interpretable and fit the observed data well. Low-rank recurrent neural networks (RNNs) exhibit such interpretability by having tractable dynamics. However, it is unclear how to best fit low-rank RNNs to data consisting of noisy observations of an underlying stochastic system. Here, we propose to fit stochastic low-rank RNNs with variational sequential Monte Carlo methods. We validate our method on several datasets consisting of both continuous and spiking neural data, where we obtain lower dimensional latent dynamics than current state of the art methods. Additionally, for low-rank models with piecewise-linear nonlinearities, we show how to efficiently identify all fixed points in polynomial rather than exponential cost in the number of units, making analysis of the inferred dynamics tractable for large RNNs. Our method both elucidates the dynamical systems underlying experimental recordings and provides a generative model whose trajectories match observed variability.

## 1 Introduction

A common goal of many scientific fields is to extract the dynamical systems underlying noisy experimental observations. In particular, in neuroscience, much work is devoted to understanding the coordinated firing of neurons as being implemented through underlying dynamical systems [1–5]. Recurrent neural networks (RNNs) constitute a common model-class of neural dynamics [6–13] which can be reverse-engineered to form hypotheses about neural computations [14, 15]. As a result, several recent research directions have centered on interpretable or analytically tractable RNN architectures. In particular, RNNs with low-rank structure [16–22] admit a direct mapping between high-dimensional population activity and an underlying low-dimensional dynamical system. RNNs with piecewise-linear activations [8, 9, 23–26] are tractable, as they have fixed points and cycles that can be accessed analytically.

To serve as useful models of brain activity, it is important that models also capture the observed brain activity, including trial-to-trial variability. Many methods that fit RNNs to data are restricted to RNNs with deterministic transitions [6–8, 10–12]. It is unlikely that, in general, all variability in the data can be explained by variability in the RNNs initial state. Thus, adopting stochastic transitions is imperative. While probabilistic sequence models are used effectively in neuroscience [27], they have so far largely consisted of state space models without an obvious mechanistic interpretation [28–32].

---

{firstname.secondname}@uni-tuebingen.de

38th Conference on Neural Information Processing Systems (NeurIPS 2024).

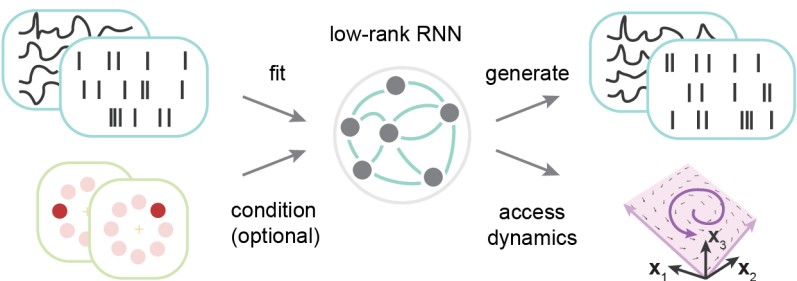

Figure 1: Our goal is to obtain generative models from which we can sample realistic neural data while having a tractable underlying dynamical system. We achieve this by fitting stochastic low-rank RNNs with variational sequential Monte Carlo.

Here, we demonstrate that we can fit large stochastic RNNs to noisy high-dimensional data. First, we show that, by combining variational sequential Monte Carlo methods [33–35] with low-rank RNNs, we can efficiently fit stochastic RNNs with many units by learning the underlying low-dimensional dynamical system. The resulting RNNs are generative models of neural data that can be used to sample trajectories of arbitrary length, and also allow for conditional generation with (both time-varying and stationary) inputs. Second, we show that, for low-rank networks with piecewise-linear activation functions, the resulting dynamics can be efficiently analyzed: In particular, we show how *all* fix points can be found with a polynomial cost in the number of units — dramatically more efficient than the exponential cost in the general case.

We first validate our method using several teacher-student setups and show that we recover both the ground truth dynamics and stochasticity. We then fit our model to several real-world datasets, spanning both spiking and continuous data, where we obtain a generative model which needs lower dimensional latent dynamics than current state of the art methods. We also demonstrate how in our low-rank RNNs fixed points can be efficiently inferred — potentially at a lower cost than approximate methods [25], while additionally coming with the guarantee that *all* fixed points are found.

## 2    Theory and methods

### 2.1    Low-rank RNNs

#### 2.1.1    Access to the low-dimensional dynamics underlying large networks

Our goal is to infer recurrent neural network models of the form

$$\tau \frac{d\mathbf{x}}{dt} = -\mathbf{x}(t) + \mathbf{J}\phi(\mathbf{x}(t)) + \Gamma_{\mathbf{x}}\xi(t), \tag{1}$$

with neuron activity $\mathbf{x}(t) \in \mathbb{R}^N$, time-constant $\tau \in \mathbb{R}_{>0}$, recurrent weights $\mathbf{J} \in \mathbb{R}^{N \times N}$, element-wise nonlinearity $\phi$, an $R$ dimensional white noise process $\xi(t)$ and $\Gamma_{\mathbf{x}} \in \mathbb{R}^{N \times R}$. In particular, we are interested in the case where the weight matrix $\mathbf{J}$ has rank $R \leq N$, i.e., it can be written as $\mathbf{J} = \mathbf{M}\mathbf{N}^{\mathsf{T}}$, with $\mathbf{M}, \mathbf{N} \in \mathbb{R}^{N \times R}$ ([18–21]). Assuming that $\mathbf{x}(0)$ lies in the subspace spanned by the columns of $\mathbf{M}$ and $\Gamma_{\mathbf{x}} = \mathbf{M}\Gamma_{\mathbf{z}}$, with $\Gamma_{\mathbf{z}} \in \mathbb{R}^{R \times R}$ , we can rewrite Eq. 1 as an equivalent $R$ dimensional system,

$$\tau \frac{d\mathbf{z}}{dt} = -\mathbf{z}(t) + \mathbf{N}^{\mathsf{T}}\phi(\mathbf{M}\mathbf{z}(t)) + \Gamma_{\mathbf{z}}\xi(t), \tag{2}$$

where we can switch between Eq. 1 and Eq. 2 by means of linear projection, $\mathbf{z}(t) = (\mathbf{M}^{\mathsf{T}}\mathbf{M})^{-1}\mathbf{M}^{\mathsf{T}}\mathbf{x}(t)$ and $\mathbf{x}(t) = \mathbf{M}\mathbf{z}(t)$. Note that we can directly extend these equations to include input, representing, e.g., experimental stimuli or context. Even if these stimuli are time-varying, $\mathbf{x}$ will be constrained to the span of the input weights and $\mathbf{M}$. By including input to the RNN, we can use the fit models for conditional generation (see Supplement C.2).

### 2.1.2 Low-rank RNNs as state space models

We consider nonlinear latent dynamical systems with observations $\mathbf{y}_t$:

$$p(\mathbf{z}_{1:T}, \mathbf{y}_{1:T}) = p(\mathbf{z}_1) \prod_{t=2}^{T} p(\mathbf{z}_t \mid \mathbf{z}_{t-1}) \prod_{t=1}^{T} p(\mathbf{y}_t \mid \mathbf{z}_t),$$

$$p(\mathbf{z}_t \mid \mathbf{z}_{t-1}) = \mathcal{N}(F(\mathbf{z}_{t-1}), \Sigma_{\mathbf{z}}), \ p(\mathbf{z}_1) = \mathcal{N}(\mu_{\mathbf{z}_1}, \Sigma_{\mathbf{z}_1}),$$

$$p(\mathbf{y}_t \mid \mathbf{z}_{t-1}) = G(\mathbf{z}_t),$$

where the transition distribution is parameterised by discretising a low-rank RNN with timestep $\Delta_t$, we have mean $F(\mathbf{z}_t) = a\mathbf{z}_t + \tilde{\mathbf{N}}^\mathsf{T}\phi(\mathbf{M}\mathbf{z}_t)$, with $a = 1 - \frac{\Delta_t}{\tau}$ and $\tilde{\mathbf{N}} = \frac{\Delta_t}{\tau}\mathbf{N}$, and covariance $\Sigma_{\mathbf{z}}$ (see Supplement C.1). The specific form of the observation function $G$, depends on the data-modality, e.g., here we use a Poisson distribution for count observations. This formulation allows one to keep the one-to-one correspondence between RNN units (or a subset of those) and recorded data neurons, (as was desired in previous work, e.g., [10–12, 36]). For example, assuming Gaussian observation noise, we can simply use that $\mathbf{x}_t = \mathbf{M}\mathbf{z}_t$ and define $G = \mathcal{N}(\mathbf{M}\mathbf{z}_t, \Sigma_{\mathbf{y}})$.

Once we learn $p(\mathbf{z}_{1:T}, \mathbf{y}_{1:T})$, we can use the obtained RNN as a generative model to sample trajectories, and reverse engineer the underlying dynamics to gain insight in the data generation process. Given the sequential structure of the RNN, we can do model learning by using variational sequential Monte Carlo (also called Particle Filtering) methods [33–35].

## 2.2 Model learning with variational sequential Monte Carlo

### 2.2.1 Sequential Monte Carlo

Sequential Monte Carlo (SMC) can be used to approximate sequences of distributions, such as those generated by our RNN, with a set of $K$ trajectories of latents $\mathbf{z}_{1:T}$ (commonly called particles) [37]. As we do not have direct access to the posterior $p(\mathbf{z}_t \mid \mathbf{y}_{1:t})$, we instead sample from a proposal distribution $r$, and adjust for the discrepancy between the proposal and target posterior distribution using importance weights. Thus, a crucial choice when doing SMC is picking the right proposal distribution $r$, from which we can sample latents conditioned on the previous latent $\mathbf{z}_{t-1}$ and observed data $\mathbf{y}_{1:T}$, or a subset of those. Given initial samples $\mathbf{z}_1^{1:K} \sim r$ and corresponding importance weights $\overline{w}_1^{1:K}$ (as defined below) SMC progresses by repeatedly executing the following steps:

$$resample \qquad a_{t-1}^k \sim \mathsf{Discrete}(a_{t-1}^k \mid \overline{w}_{t-1}^k),$$

$$propose \qquad \mathbf{z}_t^k \sim r(\mathbf{z}_t^k \mid \mathbf{y}_t, \mathbf{z}_{t-1}^{a_{t-1}^k}),$$

$$reweight \qquad w_t^k = \frac{p(\mathbf{y}_t, \mathbf{z}_t^k \mid \mathbf{z}_{t-1}^{a_{t-1}^k})}{r(\mathbf{z}_t^k \mid \mathbf{y}_t, \mathbf{z}_{t-1}^{a_{t-1}^k})},$$

with $\overline{w}_t^k = \frac{w_t^k}{\sum_{j=1}^K w_t^j}$. Here, the resampling step avoids most of the weights from concentrating on very few particles. Using SMC, we obtain, at time $t$, a filtering approximation to the posterior,

$$q_{\mathsf{filt}}(\mathbf{z}_{1:t} \mid \mathbf{y}_{1:t}) = \sum_{k=1}^K \overline{w}_t^k \delta(\mathbf{z}_{1:t}^k). \tag{3}$$

The unnormalised weights give an unbiased estimate to the marginal likelihood,

$$\hat{p}(\mathbf{y}_{1:T}) = \prod_{t=1}^T \frac{1}{K} \sum_{k=1}^K w_t^k. \tag{4}$$

We now detail how we pick the proposal distribution $r$. For linear Gaussian observations $G = \mathcal{N}(\mathbf{W}\mathbf{z}_t, \Sigma_{\mathbf{y}})$, we set $r(\mathbf{z}_t \mid \mathbf{y}_t, \mathbf{z}_{t-1}) = p(\mathbf{z}_t \mid \mathbf{y}_t, \mathbf{z}_{t-1})$, as this is available in closed form and is optimal (in the sense that it minimises the variance of the importance weights [37])

$$r(\mathbf{z}_t \mid \mathbf{y}_t, \mathbf{z}_{t-1}) = \mathcal{N}((\mathbf{I} - \mathbf{K}\mathbf{W})F(\mathbf{z}_{t-1}) + \mathbf{K}\mathbf{y}_t, \Lambda_{\mathbf{z}}) \tag{5}$$

with $\mathbf{K}$ the Kalman Gain: $\mathbf{K} = \Lambda_{\mathbf{z}}\mathbf{W}^{\mathsf{T}}\Sigma_{\mathbf{y}}^{-1}$, and $\Lambda_{\mathbf{z}} = (\Sigma_{\mathbf{z}}^{-1} + \mathbf{W}^{\mathsf{T}}\Sigma_{\mathbf{y}}^{-1}\mathbf{W})^{-1}$ (or equivalently $\mathbf{K} = \Sigma_{\mathbf{z}}\mathbf{W}^{\mathsf{T}}(\mathbf{W}\Sigma_{\mathbf{z}}\mathbf{W}^{\mathsf{T}} + \Sigma_{\mathbf{y}})^{-1}$, and $\Lambda_{\mathbf{z}} = (\mathbf{I} - \mathbf{KW})\Sigma_{\mathbf{z}}$). For non-linear observations, we can not invert the observation process in closed form, so we instead jointly optimize a parameterized 'encoding' distribution $e(\mathbf{z}_t \mid \mathbf{y}_{t-t':t})$ (as in a variational autoencoder [38]). In particular, we assume $e$ to be a multivariate normal with diagonal covariance, which we parameterize by a causal convolutional neural network, such that each latent is conditioned on the $t'$ latest observations (although sometimes non-causal encoders can be advantageous, see Supplement B.5). We then use the following proposal:

$$r(\mathbf{z}_t \mid \mathbf{z}_{t-1}, \mathbf{y}_{t-t':t}) \propto e(\mathbf{z}_t \mid \mathbf{y}_{t-t':t})p(\mathbf{z}_t \mid \mathbf{z}_{t-1}), \tag{6}$$

where we now also assume $p(\mathbf{z}_t \mid \mathbf{z}_{t-1})$ has a diagonal covariance matrix.

### 2.2.2 Relationship to Generalised Teacher Forcing

In our approach, the mean of the proposal distribution at time $t$ is a linear combination between the RNN predicted state $F(\mathbf{z}_{t-1})$ and a data-inferred state $\hat{\mathbf{z}}_t$. A recent study obtained state-of-the art results for reconstructing dynamical systems by fitting deterministic RNNs with a method called Generalised Teacher Forcing (GTF), which similarly linearly interpolates between a data-inferred and an RNN predicted state at every time-step [8]; the model propagates forward in time as $\mathbf{z}_t = (1-\alpha)F(\mathbf{z}_{t-1}) + \alpha\hat{\mathbf{z}}_t$. Hess et al. [8] showed that by choosing the appropriate $\alpha$, one can completely avoid exploding gradients, while still allowing backpropagation through time, and thus obtaining long-term stable solutions [39]. The optimal $\alpha$ can be picked based on the maximum Lyaponuv exponent of the system (a measure of how fast trajectories diverge in a chaotic system).

By including the RNN in the proposal distribution, we similarly to GTF allow backpropagation through time through the sampled trajectories. The linear combination is given by $\alpha = (\Sigma_{\mathbf{z}}^{-1} + \mathbf{W}^{\mathsf{T}}\Sigma_{\mathbf{y}}^{-1}\mathbf{W})^{-1}\mathbf{W}^{\mathsf{T}}\Sigma_{\mathbf{y}}^{-1}\mathbf{W}$ in Eq. 5, and similarly in Eq. 6 by $\alpha = (\Sigma_{\mathbf{z}}^{-1} + \Sigma_{\hat{\mathbf{z}}_t}^{-1})^{-1}\Sigma_{\hat{\mathbf{z}}_t}^{-1}$, where $\Sigma_{\hat{\mathbf{z}}_t}$ is the predicted variance of the encoding network. Thus, instead of interpolating based on an estimate of how chaotic the system is, our approach combines RNN and data inferred states adaptively (every time step, if Eq. 6 is used) based on how relatively noisy the transition distribution is with respect to the data-inferred states at time $t$, analogous to, e.g., the gain of a Kalman filter. In the formulation of GTF of Hess et al. [8], an invertable observation model is required. By learning an encoder that predicts a distribution over latents, our method naturally extends to models with non-invertible (e.g., Poisson) observations.

### 2.2.3 Variational objective

We can fit our RNNs to data by using SMC to specify a variational objective [33–35]. In variational inference, we specify a family of parameterized distributions $Q$, and optimize those parameters such that a divergence (usually the KL divergence) between the variational distribution $q(\mathbf{z}_{1:T}) \in Q$ and the true posterior $p(\mathbf{z}_{1:T} \mid \mathbf{y}_{1:T})$ is minimized. We do this by maximising a lower bound (ELBO) to the log likelihood $p(\mathbf{y}_{1:T})$. In particular, we can use Eq. 4 to specify the ELBO objective [33–35]

$$\mathcal{L} = \mathbb{E}_{q_{\mathsf{smc}}(\mathbf{z}_{1:T}^{1:K}, a_{1:T-1}^{1:K} \mid \mathbf{y}_{1:T})}[\log \hat{p}(\mathbf{y}_{1:T})], \tag{7}$$

with $q_{\mathsf{smc}}$ the sampling distribution:

$$q_{\mathsf{smc}}(\mathbf{z}_{1:T}^{1:K}, a_{1:T-1}^{1:K} \mid \mathbf{y}_{1:T}) = \prod_{k=1}^{K} r(\mathbf{z}_1^k \mid \mathbf{y}_1) \prod_{k=1}^{K} \prod_{t=2}^{T} r(\mathbf{z}_t^k \mid \mathbf{z}_{t-1}^{a_{t-1}^k} \mathbf{y}_t)\mathsf{Discrete}(a_{t-1}^k \mid \overline{w}_{t-1}^k).$$

During each training iteration, we run SMC, using the closed form optimal proposal (Eq. 5) if observations are linear Gaussian, otherwise the proposal includes a parameterised encoder (Eq. 6). We can then use the resulting unnormalised importance weights (Eq. 4) to estimate the ELBO, which we maximise with backpropagation (through time). As suggested in previous studies [33–35, 40], we use biased gradients during optimization by dropping high-variance terms arising from the resampling.

### 2.3 Finding fixed points in piecewise-linear low-rank RNNs

After having learned our model, we can gain insight into the mechanisms underlying the data generation process by reverse engineering the learned dynamics [15], e.g., by calculating their fixed points. Here, we show that the fixed points can be found analytically and efficiently for low-rank networks with piecewise-linear activation functions. This class of activation functions $\phi(\mathbf{x}_i) = \sum_d^D \mathbf{b}_i^{(d)}\mathsf{max}(\mathbf{x}_i - \mathbf{h}_i^{(d)}, 0)$ includes, e.g., the standard ReLU ($\phi(\mathbf{x}_i) = \mathsf{max}(\mathbf{x}_i - \mathbf{h}_i, 0)$) or

the 'clipped' variant ($\phi(\mathbf{x}_i) = \mathsf{max}(\mathbf{x}_i + \mathbf{h}_i, 0) - \mathsf{max}(\mathbf{x}_i, 0)$) [8] which we used in all experiments with real-world data here.

Naively, the cost of finding all fixed points piecewise-linear networks scales *exponentially* with the number of units in the networks: we would have to solve $(D+1)^N$ systems of $N$ equations [9, 24]. If networks are low rank, it is straightforward to show that we can reduce this cost to solving $(D+1)^N$ systems of $R$ equations (See Supplement A.1). In addition, however, we show that the computational cost can be greatly reduced further: One can find *all* fixed points in a cost that is *polynomial* instead of *exponential* in the number of units:

**Proposition 1.** Assume Eq. 1, with $\mathbf{J}$ of rank $R$ and piecewise-linear activations $\phi$. For fixed rank $R$ and fixed number of basis functions $D$, we can find all fixed points in the absence of noise, that is all $\mathbf{x}$ for which $\frac{d\mathbf{x}}{dt} = 0$, by solving at most $\mathcal{O}(N^R)$ linear systems of $R$ equations. *Proof.* See Supplement A.1.

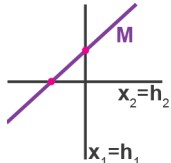

Figure 2: Proof sketch.

*Sketch.* Assuming $D = 1$, activations $\phi = \max(0, \mathbf{x}_i - \mathbf{h}_i)$; $N$ units will partition the full phase space into $2^N$ regions in which the dynamics are linear (2 units, 4 regions in Fig. 2). We can thus, in principle, solve for all fixed points by solving all corresponding linear systems of equations [9, 24]. If dynamics are confined to the $R$-dimensional subspace spanned by the columns of $\mathbf{M}$, only a subset of the linear regions (3 in Fig. 2) can be reached. Each unit partitions the space spanned by the columns of $\mathbf{M}$ with a hyperplane (pink points in Fig. 2). The amount of linear regions in $\mathbf{M}$, becomes equivalent to 'how many regions can we create in $R$-dimensional space with $N$ hyperplanes?' Using Zaslavsky's theorem [41], we can show that this at most $\sum_{r=0}^{R} \binom{N}{r} \in \mathcal{O}(N^R)$ (for fixed $R$).

## 3 Empirical Results

### 3.1 RNNs recover ground truth dynamics in student-teacher setups

We validated our method using several student-teacher setups (Fig. 3; additional statistics in Fig. S4). We first trained a 'teacher' RNN, with the weight matrix constrained to rank 2, to oscillate. We then simulated multiple trajectories with a high level of stochasticity in the latent dynamics (Fig. 3**a**, top left) and additional additive Gaussian observation noise (Fig. 3**a**, top right) on the observed neuron activity ($\mathbf{y}_i \sim \mathcal{N}(\mathbf{x}_i, \sigma_y^2)$, with $\mathbf{x} = \mathbf{Mz}$). A second 'student' RNN was then fit to the data drawn from the teacher, and both recovered the true latent dynamical system, as well as the right level of stochasticity (Fig. 3**a**, bottom; Fig. 3**d**).

We also verified that we can obtain covariance matrices $\Sigma_{\mathbf{z}}$ that are numerically close to the ground truth, for teacher networks with various levels of noise (Fig. S5). When using the bootstrap proposal (i.e., sampling from the prior; $r = p(\mathbf{z}_t \mid \mathbf{z}_{t-1})$), or too few particles, the right level of stochasticity is not obtained, indicating that the use of multiple particles and a proposal that conditions on observed data is indeed beneficial.

Given that neurons emit action potentials, which are commonly approximated as discrete events, we repeated the initial teacher-student experiment with Poisson observations generated according to $\mathbf{y}_i \sim \mathsf{Pois}(\mathsf{softplus}(\mathbf{w}_i\mathbf{x}_i - \mathbf{b}_i))$. The student RNN again recovers the oscillatory latent dynamics. Note that because of the affine transformation in the observation model, the inferred dynamics can be scaled and translated with respect to the teacher model. To verify that samples from our inferred model follow the same distribution as samples from the teacher model, we computed several statistics, which all show a close match (Fig. 3**e**; Fig. S4).

In our final teacher-student setups, we verified the ability to recover dynamics when there are known stimuli or contexts. In particular, we trained a rank-2 RNN on a task where, at each trial, it receives a transient pulse input corresponding to a particular angle $\theta$ (given as $\sin(\theta), \cos(\theta)$), and is asked to provide output matching the input after stimulus offset. The teacher RNN learns to perform the task by using an approximate ring attractor - which the student RNN accurately infers (Fig. 3**c**). Here, we inferred all fixed points by making use of Preposition 1. To demonstrate that our method also works when inputs are strongly time-varying, we included an additional setup where the teacher network was asked to report the sign of the mean of a noisy stimulus (Fig. S6).

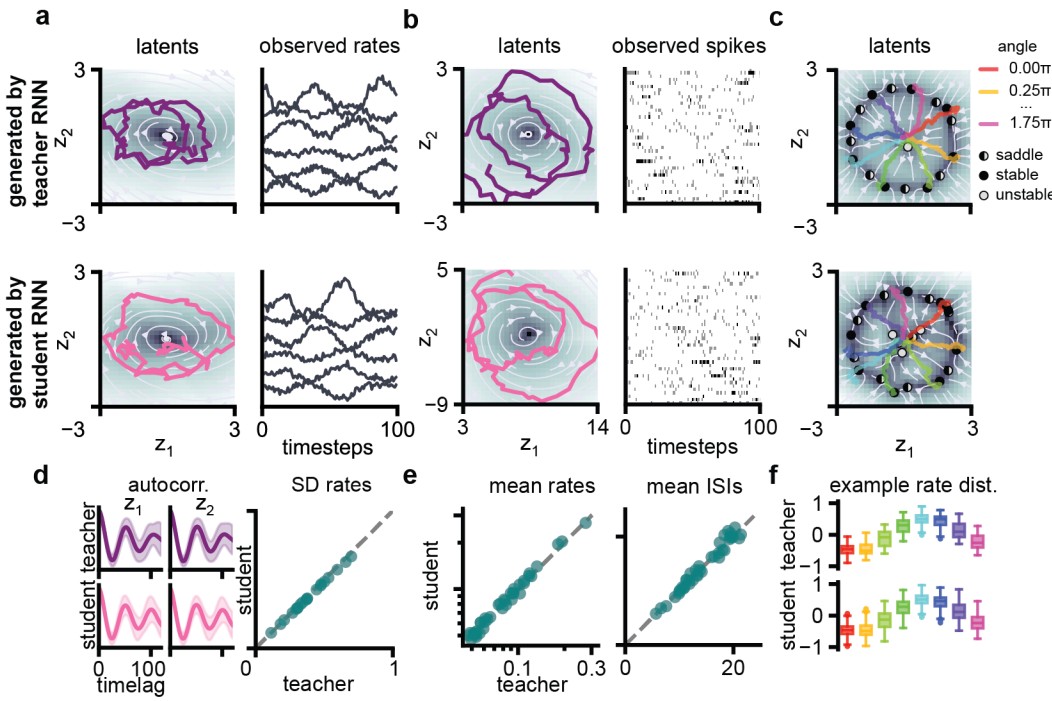

Figure 3: RNNs recover dynamics in teacher-student setups. **a)** Example ground truth latent trajectory and phase plane of low-rank RNN trained to oscillate (top left) and noisy observations of neuron activity (top right; 6/20 shown). A second low-rank RNN trained on the activity of the first recovers ground truth dynamics. **b)** Same set-up, but with Poisson observations. **c)** The teacher network was trained on a task where it has to provide an output corresponding to 8 different angles depending on an input cue. The student network, when given the same input during fitting, recovers the approximate ring attractor. **d)** Mean ($\pm$1SD) autocorrelation of the latents of the models from panel **a**, show the oscillation frequency is captured, as well as the decorrelation due to recurrent noise. The scale of the observed rates also agrees between student and teacher. **e)** Mean rates and ISI between student and teacher units of panel **b** match. **f)** Example rate distribution of one unit of the teacher and student RNN (of panel **c**), after onset of the 8 different stimuli.

## 3.2 Stochasticity allows recovering low-dimensional latents underlying EEG data

After validating our model on a toy example, we went on to several challenging real-world datasets. We first used an EEG dataset [42, 43] with 64 channels containing one minute of continuous data sampled at 160 Hz (Fig. 4). This dataset was recently used in a study where generalized teacher forcing (GTF) was used to fit deterministic RNNs with low-rank structure [8]. The GTF method obtains state-of-the-art results on several dynamical systems reconstruction tasks. It outperformed SINDy [44], neural differential equations [45], Long-Expressive-Memory [46], and other methods, while using a smaller latent dynamical system.

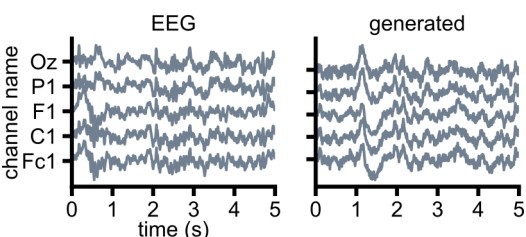

Figure 4: Example ground truth EEG [42, 43] and (unconditionally) generated traces by our model. Shown are 5/64 EEG channels.

Here we show that using a stochastic RNN with SMC instead of a deterministic RNN with GTF, we can decrease the latent dimensionality even further, from 16 to just 3 latents, while matching the original reconstruction accuracy (Table 1). We hypothesize this is because the data can be

well explained by stochastic transitions with simple underlying dynamics as opposed to complex deterministic chaos.

Table 1: Lower dimensional latent dynamics than SOTA at same sample quality. We report median ± median absolute deviation over 20 independent training runs, 'dim' refers to the dimensionality of the model's underlying dynamics and $|\theta|$ denotes the total number of *trainable* parameters. Values for GTF taken from Hess et al. [8].

| Dataset | Method | $D_{\text{stsp}} \downarrow$ | $D_H \downarrow$ | dim | $|\theta|$ |
|---|---|---|---|---|---|
| EEG (64d) | GTF [8] | $2.1 \pm 0.2$ | $0.11 \pm 0.01$ | 16 | 17952 |
| | adaptive GTF [8] | $2.4 \pm 0.2$ | $0.13 \pm 0.01$ | 16 | 17952 |
| | SMC (ours) | $2.1 \pm 0.1$ | $0.11 \pm 0.01$ | 3 | 3920 |

We evaluated samples from our RNN with two measures which were used in previous work [8], one KL divergence-based measure between the states ($D_{\text{stsp}}$), and one measure over time, based on the power spectra of generated and inferred dynamics ($D_H$; see Supplement D.3.3). Unlike Hess et al. [8], who applied smoothing, we optimized our models directly on the raw EEG data. We also fit stochastic full-rank RNNs with variational SMC, however these models tend to have worse performance on this task, while also being less interpretable (Fig. S7).

### 3.3 Interpretable latent dynamics underlying spikes recorded from rat hippocampus

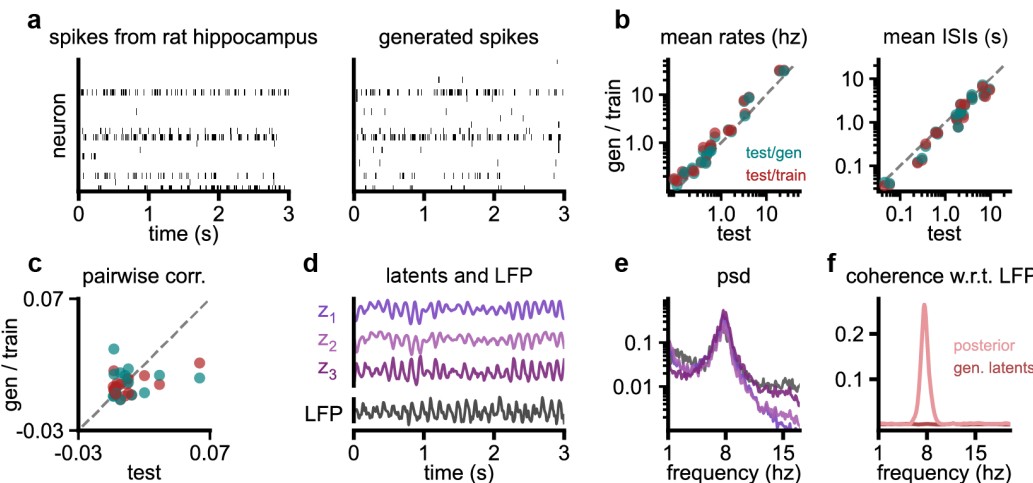

Figure 5: RNNs reproduce the stationary distribution of spiking data. **a)** We fit a rank-3 RNN to spike data recorded from rat hippocampus [47, 48] (left), and generate new samples from the RNN (right). **b)** Single neuron statistics. Mean rates and means of interspike interval (ISI) distributions of a long trajectory of data generated by the RNN (gen) match those of a held-out set of data (test). As a reference we additionally computed the same statistics between the train and test set. **c)** Population level statistics. We plot the pairwise correlations between all neurons for generated data against the pairwise correlations in the test data. **d)** The corresponding latents generated by running the RNN look visually similar to the local field potential (LFP). **e)** The peak in the power spectrum matches between latents and LFP. **f)** The posterior latents show coherence with the LFP. As a reference, we compute the coherence between the LFP and the latents generated by the RNN.

We next investigated how well our model can capture the distribution of non-continuous time series. In particular, we used publicly available electrophysiological recordings from the hippocampus of rats running to drops of water or pieces of food [47, 48]. We binned the spiking data into 10ms bins and fit a rank-3 RNN to ∼850 s of data. Samples generated by running the fit RNN autonomously closely matched the statistics of the recordings (Fig. **5a-c**). Previous investigations into this dataset have examined the relationship between spikes and theta (5-10 Hz) oscillations in the local field potential ([47]), and found that units were locked to the LFP rhythm, with the relative phase depending on the

subregions from which the units were recorded. The latents generated by the RNN are visually similar to the average local field potential (Fig. 5**d**) and match its power spectrum (Fig. 5**e**). While the model was solely trained on the spikes, the posterior latents (Eq. 3) have a clear phase relationship with the LFP, as evidenced by a high coherence between the posterior latents and LFP. In contrast, and as expected, latents from running the RNN are not correlated with the LFP (Fig. 5**f**). The correspondence between generated latents and LFP was absent when we use a related method for fitting RNNs (with deterministic transitions) to neural data (LFADS [7]; Fig. S8). Using the bootstrap proposal also led to lower-quality samples (Fig. S9).

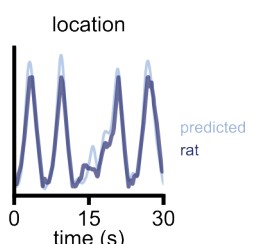

Units in rat hippocampus have been shown to code for position, e.g., through place cells [49], which tend to fire if the animal is at a specific location. To further investigate how well we can model recordings from the hippocampus, we fit a rank-4 RNN to an additional set of recordings of rats running on a linear track [50–52] (Fig. S10). As in Zhou and Wei [53], we first focus only on the spikes recorded while the rat is moving, which we bin into 25 ms bins. The RNN again accurately reconstructs the distribution of spikes and again has latent oscillations. Here the frequency at which power peaks is slightly higher than that of the LFP, potentially related to phase precession ([54]). While solely trained on spikes, the posterior latents also allowed us to predict the position of the rats with reasonable accuracy ($R^2 = 0.79 \pm 0.05$ mean $\pm$ SD, $N = 4$ RNNs; Fig. 6). We also fit rank-12 RNNs to around 15 minutes of recording (again with 25 ms bins), which includes long intermediate periods where the rat is stationary. Here our generative model learns to have higher theta power during running bouts, in line with the data (Fig. S11).

Figure 6: Posterior latents of our model (fit solely spikes) can be used to predict rat position.

## 3.4 Extracting stimulus-conditioned dynamics in monkey reaching task

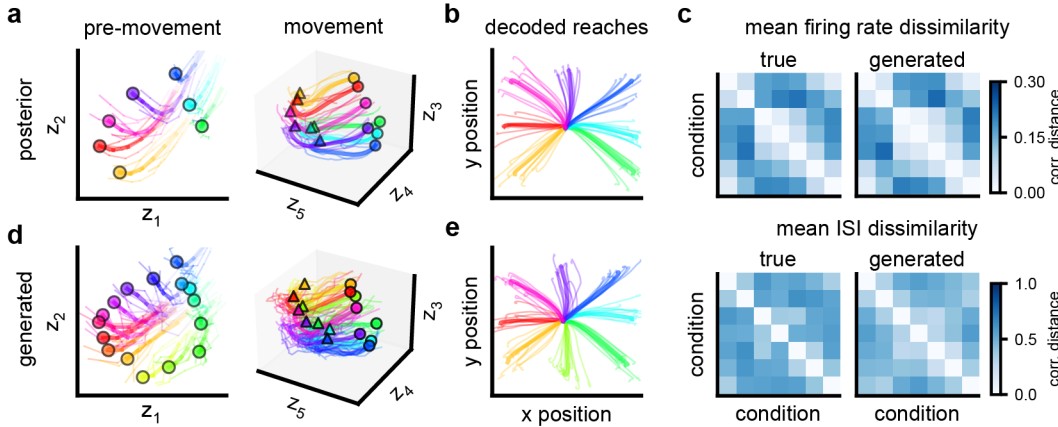

Figure 7: Inferred and generated dynamics from the model fit to macaque spiking activity during a reaching task. **a)** Latent states inferred from the macaque spiking data prior to movement initiation ('pre-movement') and during movement execution ('movement'), colored by the intended reach target. **b)** Reach trajectories decoded from model-inferred neural activity. **c)** Dissimilarity matrices computed across the seven conditions (i.e., the seven colors in **a**, **b**) for per-neuron mean firing rate and ISI. We generate neural activity from the model by providing the same conditioning stimuli as in the real data. Then, for each statistic, we compute and show the correlation distance between conditions in the real data (left) and model-generated data (right). **d**, **e)** Same as **a**, **b**, but with latent activity and behavioral predictions generated from the model with conditioning inputs including directions not seen in the real data (e.g., lime green). For clarity, we show only a subset of conditions in the decoded reaches.

We further investigated how well we can recover stimulus-conditioned dynamics. We applied our method to spiking activity recorded from the motor and premotor cortices of a macaque performing a delayed reaching task. This type of data has been popular for investigating neural dynamics underlying the control of movement [2, 3] and evaluating neuroscientific latent variable models [7, 55, 56]. We first validated the ability of our method to obtain a sensible posterior by evaluating it on the Neural Latents Benchmark [56] (Supplement B.5, Table S2).

We then went on to a set-up where we explicitly conditioned our model on external context. For simplicity, we constrained our experiment to trials with straight reach trajectories in the data. We fit a rank-5 model to these data while conditioning the RNN dynamics on the target position by providing the target position as input. Our model was able to infer single-trial latent dynamics and neuron firing rates that predict reach velocity with high accuracy at lower latent dimensionalities than models without inputs (Fig. 7**b**, $R^2 = 0.90$ for this model, see Table S3 for additional statistics).

We examined the posterior latents inferred by the model and found that our model recovers structured and interpretable latent dynamics. Before movement onset, latent states corresponded to the intended reach targets, which were near the edges of a rectangular screen (Fig. 7**a**, left), in line with [55]. During the movement period, the latents followed parallel curved trajectories that preserve target information (Fig. 7**a**, right) and can be decoded to predict monkey reach behavior (Fig. 7**b**).

We then generated neural data from the RNN conditioned on stimulus input. Again, the distribution of spikes is well-captured (Fig. S12). We additionally evaluated whether the model faithfully captures differences in spiking statistics across the seven reach directions, finding reasonable correspondence in dissimilarities between conditions in the generated and the real data (Fig. 7**c**). Finally, we simulated our trained RNN with conditioning inputs, including reach directions not present in the data, and found that the structured latent space recovered by the model enables realistic generalization to unseen reach conditions (Fig. 7**d**, **e**, lime green condition).

## 3.5   Searching for fixed points

In Proposition 1, we derived a bound on the number of systems of equations one has to solve in order to find *all* fixed points in piecewise-linear low-rank RNNs. Recently, an approximate algorithm for finding fixed points in piecewise-linear networks was proposed [25]. Here, we perform an exploration into how this compares to our analytic method by searching for fixed points of the RNN in Fig. 3**c** (top). For the same number of matrix inverses computed by our analytic method, the approximate method generally does not find all 17 fixed points (Fig. 8). We note, however, that (unlike ours) the convergence of the approximate method depends on the dynamics of the RNN, and as a result, there are theoretical scenarios where the approximate method can be shown to be faster. Yet we empirically also found scenarios where the approximate methods failed to converge within the time-frame of our experiments (Fig. S13).

Our analytic method relies on the insight that only a subset of all linear subregions formed by the piecewise-linear activations can be reached in low-rank networks. For networks with moderate rank, the cost of searching through all of the subregions might still be too high. We can, however, hugely reduce the search space of the approximate method [25] (from $(D + 1)^N$ to $\sum_{r=0}^{R} D^r \binom{N}{r}$), at an upfront cost (Supplement B.7; orange line in Fig. 8).

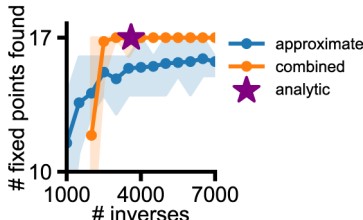

Figure 8: Comparison of our analytic method (star) and the approximate method proposed in Eisenmann et al. [25] (blue) for finding the fixed points of the teacher RNN in Fig. 3**c**. We can also use Proposition 1 to constrain the search space of the approximate method (orange). Error bars denote the minimum and maximum amount of fixed points found over 20 independent runs of the algorithm.

## 4 Discussion

Here we proposed to fit low-rank RNNs to neural data using variational sequential Monte Carlo. The resulting RNNs are generative models with tractable underlying dynamics, from which we can sample long, stable trajectories of realistic data. We validated our method on several teacher-student setups and demonstrated the effectiveness of our method on multiple challenging real-world examples, where we generally needed a latent dynamical system with very few dimensions to accurately model the data. Besides our empirical results, we obtained a theoretical bound on the cost of finding fixed points for RNNs with piecewise-linear activation functions when they are also low-rank.

Adding stochastic transitions to low-rank RNNs can potentially hugely reduce the rank required to accurately model observed data, as demonstrated here with a network fit to EEG data where we could reduce the dimensionality from 16 to just 3. While many methods that fit RNNs to neural data (e.g., [6–8, 10–12]) assume deterministic transitions, there is a rich literature concentrating on probabilistic sequence models in neuroscience (e.g., [28–32]). In particular, a recent work termed FINDR [31] uses variational inference (but not SMC), to similarly find very low-dimensional dynamical systems underlying neural data. These stochastic dynamical systems were parameterized using neural differential equations [45]. While Eq. 2 can be seen as a neural differential equation with one hidden layer, our particular formulation allows us to find its fixed-points effectively and map back to a regular, mechanistically interpretable RNN (Eq. 1) after fitting, which enables additional investigations into neural population dynamics [18, 20–22].

We here — similar to FINDR (and [57]) — did not use the adjoint method as is typical in the neural differential equation literature, but rather a simple Euler-Maruyama discretisation scheme and standard backpropagation through time. However, one could investigate how we can integrate our approach with variational approaches that use adjoint methods when fitting latent neural SDEs [58, 59] as well as with filtering approaches for continuous time systems [60]. This could be especially relevant for irregularly sampled time-series.

The reason we can do the mapping between a low-rank RNN (Eq. 1) and a latent dynamical system (Eq. 2) crucially relies on our assumption that samples from the recurrent noise process are correlated, such that they lie within the column-space of $\mathbf{M}$. Valente et al. [61] showed that for linear low-rank RNNs arbitrary covariances in the full $N$ dimensional space can be used, when increasing the dimensionality of the latent dynamics to twice the rank $R$ (to the column space of both $\mathbf{M}$ and $\mathbf{N}$), this however does not generalise to our non-linear setting. We do expect correlated recurrent noise to be appropriate for modeling stochasticity arising from unobserved inputs or from partial observations [61] —additionally, correlated noise constituted a pragmatic choice that allows building an *stochastic* model that can allow for trial-by-trial variability while maintaining the tractability of low-rank deterministic RNNs.

Still, future work can investigate training networks with more relaxed assumptions on the recurrent noise models, including extensions to non-Gaussian noise-processes. The latter could be of particular interest if more biologically plausible (i.e., spiking) neurons were used in the recurrence [36, 62].

Our results also open up further avenues to explore questions in neuroscience. The relation between LFP and spike (phase) in the hippocampus has been of great interest [47, 54, 63, 64]. While we performed some preliminary investigation into the relation between the inferred latents and the local field potential, further studies could perform a systematic investigation into their relation, for instance, by using a multi-modal setup [13], or to investigate multi-region temporal relationships and interactions [10].

Taken together, by inferring low-rank RNNs with variational SMC, we obtained generative models of neural data whose trajectories match observed variability, and whose underlying latent dynamics are tractable.

## Code availability

Code to reproduce our results is available at `https://github.com/mackelab/smc_rnns`.

## Acknowledgments

This work was supported by the German Research Foundation (DFG) through Germany's Excellence Strategy (EXC-Number 2064/1, PN 390727645), SFB 1089 (PN 227953431) and SPP2041 (PN 34721065), the German Federal Ministry of Education and Research (Tübingen AI Center, FKZ: 01IS18039; DeepHumanVision, FKZ: 031L0197B), and the European Union (ERC, DeepCoMechTome, 101089288), the 'Certification and Foundations of Safe Machine Learning Systems in Healthcare' project funded by the Carl Zeiss Foundation. MP and MG are members of the International Max Planck Research School for Intelligent Systems (IMPRS-IS). We thank Cornelius Schröder for feedback on the manuscript, and all members of Mackelab for discussions throughout the project.

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

# A  Supplemental material

## A.1  Proof of preposition 1

### A.1.1  Problem definition

We are interested in finding all fixed points of the following equation:

$$\tau \frac{d\mathbf{x}}{dt} = -\mathbf{x}(t) + \mathbf{J}\phi(\mathbf{x}(t)), \tag{8}$$

with $\mathbf{x}(t) \in \mathbb{R}^N$, element-wise nonlinearity $\phi(\mathbf{x}_i) = \sum_d^D \mathbf{b}_i^{(d)} \mathsf{max}(\mathbf{x}_i - \mathbf{h}_i^{(d)})$ and low-rank matrix $\mathbf{J} = \mathbf{M}\mathbf{N}^\mathsf{T}$, with $\mathbf{M}, \mathbf{N} \in \mathbb{R}^{N \times R}$ and $R \leq N$. Since $\tau$ only scales the speed of the dynamics, we will, for convenience and without loss of generality, assume $\tau = 1$.

### A.1.2  Preliminaries: Fixed points in Piecewise-linear RNNs

First, we briefly repeat results from Durstewitz [9]. Assume $D = 1$, $\phi(\mathbf{x}_i) = \mathsf{max}(\mathbf{x}_i - \mathbf{h}_i)$. To find all fixed points of Eq. 8, start by redefining $\phi$ by introducing a diagonal indicator matrix:

$$\mathbf{D}_\Omega = \begin{bmatrix} d_1 & & & \\ & d_2 & & \\ & & \ddots & \\ & & & d_N \end{bmatrix}, \tag{9}$$

with $d_i = \begin{cases} 1, & \text{if } \mathbf{x}_i > \mathbf{h}_i \\ 0, & \text{otherwise} \end{cases}$.

Then our RNN equation, for a given $\mathbf{x}$ and corresponding $\mathbf{D}_\Omega$ reads:

$$\frac{d\mathbf{x}}{dt} = -\mathbf{x}(t) + \mathbf{J}\mathbf{D}_\Omega\mathbf{x}(t) - \mathbf{J}\mathbf{D}_\Omega\mathbf{h}.$$

Each of the $2^N$ configurations of $\mathbf{D}_\Omega$ corresponds to a region in which the dynamics are linear. Thus, for each configuration, we can solve:

$$\mathbf{0} = -\mathbf{x} + \mathbf{J}\mathbf{D}_\Omega\mathbf{x} - \mathbf{J}\mathbf{D}_\Omega\mathbf{h},$$
$$\mathbf{x}^* = (\mathbf{J}\mathbf{D}_\Omega - \mathbf{I})^{-1}\mathbf{J}\mathbf{D}_\Omega\mathbf{h}.$$

Next, we check whether the obtained $\mathbf{x}^*$ is consistent with the assumed $\mathbf{D}_\Omega$ (Eq. 9). If so, we found a fixed point of the RNN. We have to check, as the solution to the system of linear equations can lie outside of the linear regions specified by $\mathbf{D}_\Omega$. Note that if for some $\mathbf{D}_\Omega$ the matrix $\mathbf{J}\mathbf{D}_\Omega - \mathbf{I}$ is not invertible, then there is no single fixed point, but we still can find a structure of interest (e.g., a direction with eigenvalue 0 corresponds to marginal stability, i.e., a line attractor).

## A.2  Preliminaries: Fixed points in Piecewise-linear low-rank RNNs

First, assume $\mathbf{x}(0)$ is in the subspace spanned by the columns of $\mathbf{M}$. With the low-rank assumption, we can rewrite Eq. 8 for all $t \in [0, \infty)$, by projecting it on $\mathbf{M}$ [18, 20, 21]:

$$\frac{d\mathbf{z}}{dt} = -\mathbf{z}(t) + \mathbf{N}^\mathsf{T}\phi(\mathbf{M}\mathbf{z}(t) - \mathbf{h}) \tag{10}$$

with $\mathbf{x}(t) = \mathbf{M}\mathbf{z}(t)$.

Now assume $\mathbf{x}(0)$ contains some part $\mathbf{x}^\perp(0)$ not in the subspace spanned by $\mathbf{M}$, i.e., we have $\mathbf{x}(0) = \mathbf{M}\mathbf{z}(0) + \mathbf{x}^\perp(0)$. The dynamics of $\mathbf{x}^\perp$(t) are simply given by $\frac{\mathbf{x}^\perp}{dt} = -\mathbf{x}^\perp(t)$ which will decay to its stable point at $\mathbf{0}$ irrespective of $\mathbf{z}(t)$, and can thus not contribute additional fixed points.

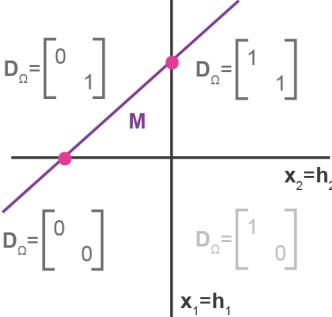

Supplementary Figure 1: Proof sketch including $\mathbf{D}_\Omega$'s. The phase-space of an RNN with $N$ (here 2) units with activation $\max(0, \mathbf{x}_i - \mathbf{h}_i)$ is partitioned into $2^N$ (here 4) regions in which the dynamics are linear, each corresponding to a configuration of $\mathbf{D}_\Omega$. If dynamics are confined to the $R$-dimensional subspace spanned by the columns of $\mathbf{M}$, only a subset (here 3) can be reached. Each unit intersects the space spanned by the columns of $\mathbf{M}$ with a hyperplane (the pink points in the Figure). The amount of linear regions in $\mathbf{M}$, thus becomes equivalent to "how many regions can we create in $R$-dimensional space with $N$ hyperplanes?"

Naively, using the same strategy as before to obtain all fixed points $\mathbf{z}$, we would need to solve $2^N$ linear systems of $R$ equations (again for all configurations of $\mathbf{D}_\Omega$):

$$\mathbf{z}^* = (\mathbf{N}^\mathsf{T}\mathbf{D}_\Omega\mathbf{M} - \mathbf{I})^{-1}\mathbf{N}^\mathsf{T}\mathbf{D}_\Omega\mathbf{h}, \tag{11}$$

### A.3   Preliminaries: Hyperplane arrangements

In the subsequent section, we will turn to the question of how many equations we need to solve to find all possible fixed points. Recall that it is possible to calculate the fixed points analytically because piecewise-linear nonlinearities partition space into subregions in which dynamics are linear. Each of the linear regions corresponds to a configuration of $\mathbf{D}_\Omega$. For networks with low-rank connectivity, we have to consider only a small subset of those, as only a small subset of all configurations of $\mathbf{D}_\Omega$ correspond to $\mathbf{x}$'s within the column space of $\mathbf{M}$ (See Fig. S1). To find out exactly how many regions lie within the column space, we will need to answer the question: *in how many regions can we divide $R$-dimensional space, with $N$ hyperplanes?* To answer this question in general, we will need a theorem from the field of *hyperplane arrangements* [41, 65–67]. Here we give a brief introduction.

**Introduction to hyperplane arrangements:**   A finite arrangements of hyperplanes is a set of $N$ affine subspaces $\mathcal{A} = \{a_1, \ldots, a_N\}$ in some vector space $V = \mathbb{R}^R$. Recall a hyperplane is an $R - 1$ dimensional subspace defined by a linear equation $a_i := \{\mathbf{v} \in V | \mathbf{m}^\mathsf{T}\mathbf{v} = h\}$ for some $\mathbf{m} \in V, h \in \mathbb{R}$. Note that any linear system of equations $\mathbf{M}\mathbf{v} = \mathbf{h}$ with $\mathbf{M} \in \mathbb{R}^{N \times R}$ equivalents defines an arrangement of $N$ hyperplanes in $R$ dimensional space. In Fig. S2**a,c**, we show arrangements of 3 hyperplanes in $\mathbb{R}^2$. In this case, a hyperplane is a line, but there are infinitely many possibilities on how we can arrange these lines in two-dimensional space. We are interested in

$$\mathcal{N}(\mathcal{A}) := \text{ number of regions } \mathcal{A} \text{ partitions } \mathbb{R}^R,$$

where regions correspond to the connected components of $\mathbb{R}^R \setminus \mathcal{A}$. In this simple case, we can visually verify that the arrangements in Fig. S2**a** partitions the space into 7 regions, whereas the arrangement in Fig. S2**c** partitions the space into only 6 regions. Clearly, the number of regions $\mathcal{A}$ partitions space in is strongly related to the number of unique intersections of lines. We have fewer regions in Fig. S2**c**, simply because all lines intersect at the same point. If we can wiggle the hyperplanes a little, and not change the number of regions (as we can do in Fig. S2**a**, but not Fig. S2**c**), we call the hyperplanes in *general* position (see Theorem 1 for a formal definition).

To count the amount of regions for any arrangement of hyperplanes, we can leverage an algebraic construction called the *intersections poset* $\mathcal{L}(\mathcal{A})$. This is the set of all nonempty intersections of hyperplanes in $\mathcal{A}$ and includes $V$. Elements of this set are generally referred to as *flats*. The flats

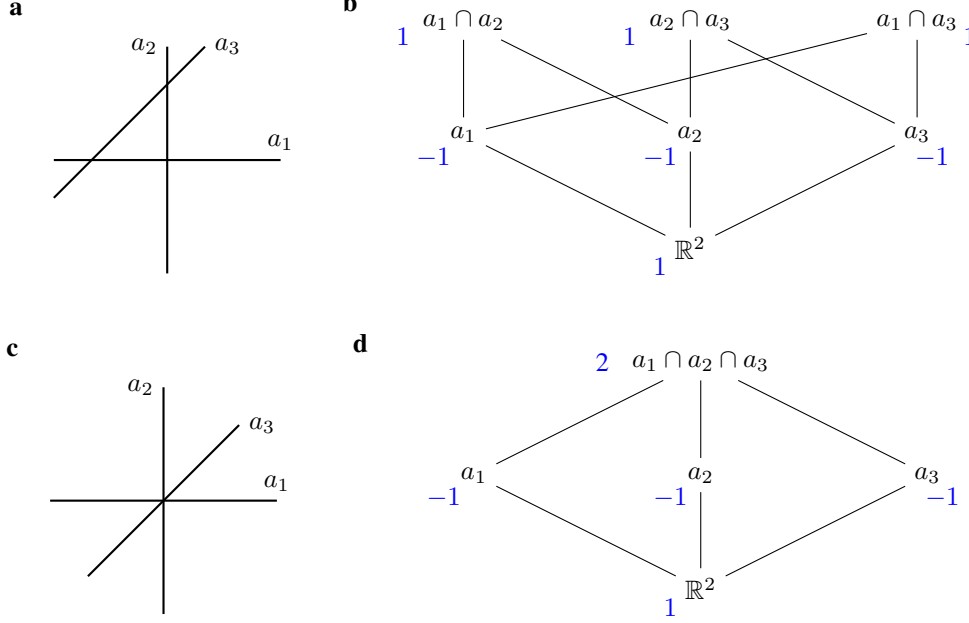

Supplementary Figure 2: **a)** An arrangement of 3 hyperplanes $a_1, a_2$ and $a_3$ in *general position.* **b)** the associated intersection poset of the arrangement in **a**. **c)** An alternative arrangement with its associated intersection poset. **d)**. Blue numbers indicate the value of the Möbius function.

are ordered by reverse inclusion $x \leq y \iff x \supseteq y$ in the intersection poset. We visualized example intersection posets of the previous examples (Fig. S2**b, d**). Here we organized the flats by dimensionality (such a visualization is called a Hesse Diagram). Importantly for any real arrangement $\mathcal{A}, \mathcal{N}(\mathcal{A})$ solely depends on $\mathcal{L}(\mathcal{A})$ (Corollary 2.1, [67]).

To calculate $\mathcal{N}(\mathcal{A})$ from $L(\mathcal{A})$, we need one last construction, namely the Möbius function, recursively defined by

$$\mu(\mathcal{X}, s) = \begin{cases} 1 & \text{if } s = \mathcal{X} \\ -\sum_{\mathcal{X} \supseteq s' \supset s} \mu(\mathcal{X}, s'), & \text{if } s \subset \mathcal{X}. \end{cases} \tag{12}$$

The numerical values for the example are shown in Fig. S2.

**Theorem 1** (Zaslavsky's Theorem; [41, 67]). *Given a vector space $V = \mathbb{R}^R$ and an arrangement of $N$ hyperplanes $\mathcal{A} = \{a_1, \ldots, a_N\}$ on $V$, then the number of regions $\mathcal{A}$ partitions $V$ in (denoted $\mathcal{N}(\mathcal{A})$, can be expressed as follows*

$$\mathcal{N}(\mathcal{A}) = \sum_{s \in L(\mathcal{A})} \mu(\mathbb{R}^R, s)(-1)^{\dim(s)}$$

*furthermore, it holds that*

$$\mathcal{N}(\mathcal{A}) \leq \sum_{r=0}^{R} \binom{N}{r} \tag{13}$$

*with equality if and only if $\mathcal{A}$ is in general position i.e., $\mathcal{A}$ must satisfy*

*(i)* $\{a_1, \ldots, a_p\} \subseteq \mathcal{A}$ *and* $p \leq R \Rightarrow \dim(\bigcap_{i=1}^{p} a_i) = N - p$

*(ii)* $\{a_1, \ldots, a_p\} \subseteq \mathcal{A}$ *and* $p > R \Rightarrow \bigcap_{i=1}^{p} a_i = \emptyset$

One can verify this fact for the given example shown in Fig. S2. We refer to Stanley [67] for an in-depth formal introduction to this topic. Fundamentally, it is based on the following recursion that the number of regions for any arrangement satisfies

$$\mathcal{N}(\mathcal{A} \cup \{a_{N+1}\}) = \mathcal{N}(\mathcal{A}) + \mathcal{N}(\mathcal{A}^{a_{N+1}})$$

where $\mathcal{A}^{a_{N+1}} := \{a_{N+1} \cap a_i | a_i \in \mathcal{A}, a_{N+1} \cap a_i \neq \emptyset, a_{N+1} \not\subseteq a_i\}$ (Lemma 2.1, Stanley [67]). Note that $a_{N+1}$ is itself an $R-1$ dimensional vector space, and each intersection $a_{N+1} \cap a_i$ is an $R-2$ dimensional hyperplane within $a_{N+1}$ (e.g., the intersection of two planes is a line within the planes). Hence $\mathcal{A}^{a_{N+1}}$ is itself an arrangement of $N$ hyperplanes, but in an $R-1$ dimensional subspace. In fact, the intersection poset exhaustively enumerates the elements of all possible $\mathcal{A}^{a_i}$, and the Möbius function can be shown to satisfy the above recursion.

If we choose $\phi(\mathbf{x}_i) = \max(\mathbf{x}_i - \mathbf{h}_i, 0)$ i.e $D = 1$, each neuron would partition space by a single hyperplane $\mathbf{x}_i = \mathbf{h}_i$ or equivalently the $R$ dimensional subspace by the hyperplane $\mathbf{M}_i \mathbf{z} = \mathbf{h}_i$. Hence, the hyperplane arrangement is determined by the matrix $\mathbf{M}$ and offset $\mathbf{h}$. As these quantities are learned during training of the RNN, this arrangement is often in a general position because it is simply numerically unlikely that two hyperplanes are exactly parallel or intersect in exactly the same "point".This does, however, change in the general case $D > 1$, for which we derive a tighter bound in the section below.

**Arrangements of parallel families**    For the general case $\phi(\mathbf{x}_i) = \sum_{d=1}^{D} \mathbf{b}_i^{(d)} \max(\mathbf{x}_i - \mathbf{h}_i^{(d)}, 0)$ each neuron will partition space with $D$ hyperplanes $\mathbf{b}_i^{(d)} \mathbf{x}_i = \mathbf{b}_i^{(d)} \mathbf{h}_i^{(d)} \iff \mathbf{x}_i = \mathbf{h}_i^{(d)}$ as before; equivalently each neuron partitions the $R$ dimensional subspace with $D$ hyperplanes $\mathbf{M}_i \mathbf{z} = \mathbf{h}_i^{(d)}$. Notably, all the $D$ hyperplanes here will share the same row of $\mathbf{M}$, and thus they are *parallel*. Clearly, any such arrangement cannot be in general arrangement by definition.

The resulting arrangement will have a very specific structure. Let's define

$$A_i := \{a_{i1}, \dots, a_{iD}\}$$

as a *family* of $D$ parallel hyperplanes. Any pair of hyperplanes $a_{il}, a_{im} \in A_i$ is parallel. A low-rank RNN with $N$ neurons and a general piecewise-linear activation function will thus lead to an arrangement consisting of $N$ families of $D$ parallel hyperplanes.

We can use this specific structure to obtain a tighter bound.

**Lemma 1.** *Let $\mathcal{A} = A_1 \cup \cdots \cup A_{N-1}$ be an arrangement of $N-1$ families of $D$ parallel lines, then it satisfies the following recursion*

$$\mathcal{N}(\mathcal{A} \cup A_N) = \mathcal{N}(\mathcal{A}) + \sum_{d=1}^{D} \mathcal{N}\left(\mathcal{A}^{a_{Nd}}\right).$$

*Furthermore, denote by $\mathcal{N}(N, R, D)$ the maximum number of regions attainable by any arrangement of $N$ families of $D$ parallel hyperplanes in $R$ dimensional space then*

$$\mathcal{N}(N, R, D) \leq \mathcal{N}(N-1, R, D) + D \cdot \mathcal{N}(N-1, R-1, D).$$

*Proof.* To add $A_N$ to $\mathcal{A}$, we have to add $D$ new parallel hyperplanes. We can do so by iteratively applying Lemma 2.1 [67]. We obtain

$$\mathcal{N}(\mathcal{A} \cup \{a_{N1}, \dots, a_{ND}\}) = \mathcal{N}(\mathcal{A} \cup \{a_{N1}, \dots, a_{N(D-1)}\}) + \mathcal{N}\left((\mathcal{A} \cup \{a_{N1}, \dots, a_{N(D-1)}\})^{a_{ND}}\right)$$

$$= \mathcal{N}(\mathcal{A}) + \sum_{d=1}^{D} \mathcal{N}\left(\left[\mathcal{A} \cup \bigcup_{i=1}^{d-1} \{a_{Ni}\}\right]^{a_{Nd}}\right).$$

Now note that $\mathcal{A}^{a_{Nj}} := \{a_{Nj} \cap a_{lm} | a_{lm} \in \mathcal{A}, a_{Nj} \cap a_{lm} \neq \emptyset, a_{Nj} \not\subseteq a_{lm}\}$, hence by definition only hyperplanes that intersect with $a_{Nj}$ are included in this set. As $a_{Nj}$ is parallel to any other $a_{Ni}$ for all $i \neq j$, all $a_{Nj} \cap a_{Ni}$ cannot be in the set. Hence for any $d$, we have that

$$\mathcal{N}\left(\left[\mathcal{A} \cup \bigcup_{i=1}^{d-1} \{a_{Ni}\}\right]^{a_{Nd}}\right) = \mathcal{N}(\mathcal{A}^{a_{Nd}})$$

which proves the first equation.

Recall that we define $\mathcal{N}(N, R, D)$ as the maximum number of regions attainable by any arrangement. Notice that $\mathcal{A}$ by construction is an arrangement of $N-1$ families of $D$ parallel hyperplanes in $R$ dimension. Thus by definition $\mathcal{N}(\mathcal{A}) \leq \mathcal{N}(N-1, R, D)$.

As noted before, the intersection set of two hyperplanes in dimension $R$ is itself a hyperplane of dimension $R - 1$. Furthermore the intersection sets of $D$ parallel hyperplanes with $a_{Nd}$, remain parallel. Hence $\mathcal{A}^{a_{Nd}}$ is an arrangement of at most $N - 1$ families of $D$ parallel hyperplanes in $R - 1$ dimensions. Thus $\mathcal{N}(\mathcal{A}^{a_{Nd}}) \leq \mathcal{N}(N - 1, R - 1, D)$ leaving us with

$$\mathcal{N}(\mathcal{A} \cup \{a_{N1}, \ldots, a_{ND}\}) \leq \mathcal{N}(N - 1, R, D) + D \cdot \mathcal{N}(N - 1, R - 1, D).$$

As this holds for any arrangement, it also holds for the arrangement that has $\mathcal{N}(N, R, D)$ regions (i.e., which maximizes the number of regions) and, therefore, proves the second equation.

$\square$

**Lemma 2.** *Let $\mathcal{A}$ be an arrangement of $N$ families of $D$ parallel hyperplanes. Then, it holds that*

$$N(\mathcal{A}) \leq \sum_{r=0}^{R} D^r \binom{N}{r}$$

*with equality if each family is in a general position, i.e., that every subarrangement $\{a_{1j_1}, \ldots, a_{Nj_N}\}$ for all $1 \leq j_i \leq D$ is in general position.*

*Proof.* We will first construct an intersection poset $L(\mathcal{A})$ on the level of families $A_i$ in general position. After all, the *intersection properties* between these families is the same as between their elements, e.g., if $a_{i1}$ intersects $a_{j1}$ then also all lines in $A_i$ intersect all lines in $A_j$.

The resulting intersection poset $L(\mathcal{A})$ can be clustered into the corresponding families. We visualize the construction in Fig. S3.

At each rank $r$ (level from bottom to top), we can choose exactly $\binom{N}{r}$ families of hyperplanes that intersect (exactly the case if we just have $N$ hyperplanes in general position). To obtain a flat of dimension $R - r$ we have to choose $r$ out of the $N$ hyperplane families without replacement.

If, e.g., two families of parallel hyperplanes $A_i, A_j$ intersect, then any element $a_{ik}$ will intersect with any element $a_{jl}$ for all $1 \leq k, l \leq D$ leading to at most $D^2$ flats within each family (there can be less as other families might intersect in the same "point"). In general, each cluster of intersections of $r$ families will contain at most $D^r$ flats.

By construction of $L(\mathcal{A})$ and Theorem 1, the lemma follows directly.

To show that this construction indeed is an upper bound for all arrangements, we can use Lemma 1. There, we established a recursion, which any such upper bound must satisfy. Hence, assume $\mathcal{N}(N, R, D) = \sum_{r=0}^{R} D^r \binom{N}{r}$. Notice that using Pascal's identity, we can rewrite

$$\begin{aligned}
\mathcal{N}(N, R, D) &= \sum_{r=0}^{R} D^r \binom{N}{r} \\
&= \sum_{r=0}^{R} D^r \left( \binom{N-1}{r} + \binom{N-1}{r-1} \right) \\
&= \sum_{r=0}^{R} D^r \binom{N-1}{r} + \sum_{r=0}^{R} D^r \binom{N-1}{r-1} \\
&= \sum_{r=0}^{R} D^r \binom{N-1}{r} + \underbrace{D^0 \binom{N-1}{-1}}_{:=0} + \sum_{r=1}^{R} D^r \binom{N-1}{r-1} \\
&= \mathcal{N}(N-1, R, D) + \sum_{r=0}^{R-1} D^{r+1} \binom{N-1}{r} \\
&= \mathcal{N}(N-1, R, D) + D \cdot \mathcal{N}(N-1, R-1, D)
\end{aligned}$$

$\square$

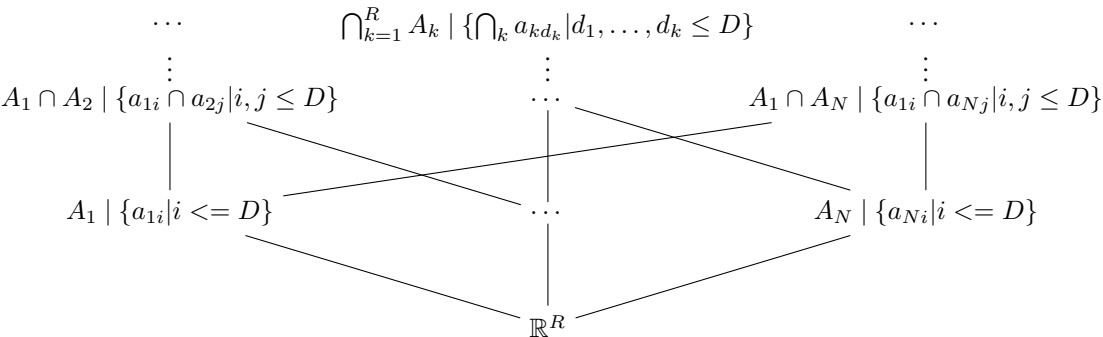

Supplementary Figure 3: Construction of the intersection poset $L(\mathcal{A})$ for an arrangement of $N$ families $A_i$ of $D$ parallel hyperplanes in "general position".

## A.4 Proof of proposition

Using the previously derived techniques, we will prove here the main proposition. Furthermore, in Algorithm 1, pseudo-code is given to compute all fixed points in practice.

**Proposition 1.** *Assume the RNN of Eq. 8, with $\mathbf{J}$ of rank $R$ and piecewise-linear activations: $\phi(\mathbf{x}_i) = \sum_d^D \mathbf{b}_i^{(d)} \max(\mathbf{x}_i - \mathbf{h}_i^{(d)}, 0)$. For fixed rank $R$ and fixed number of basis functions $D$, we can find all fixed points in the absence of noise, that is all $\mathbf{x}$ for which $\frac{d\mathbf{x}}{dt} = 0$, by solving at most $\mathcal{O}(N^R)$ linear systems of $R$ equations (for fixed $R$).*

*Proof.* By definition, each neuron partitions $\mathbb{R}^N$ in $D+1$ linear regions with $D$ hyperplanes described by $\mathbf{x}_i^{(d)} = \mathbf{h}_i^{(d)}$, for the $i$'th neuron. Using that in the columnspace of $\mathbf{M}$, we have $\mathbf{x} = \mathbf{Mz}$, it follows that each neuron partitions the $R$ dimensional subspace spanned by columns of $\mathbf{M}$, with $D$ hyperplanes described by $\sum_r^R \mathbf{M}_{i,r} \mathbf{z}_r = \mathbf{h}_i^{(d)}$. Notice that these hyperplanes are parallel, as they all share the same coefficients $\mathbf{M}_i$ but have a different offset $\mathbf{h}_i^{(d)}$. Using Lemma 2 we know that there can only be $\sum_{r=0}^R D^r \binom{N}{r}$ such regions.

How do we find those regions? Let's first consider the case of $D = 1$, and assume that the hyperplanes are in general position. We can find the corresponding configurations of $\mathbf{D}_\Omega$ as follows. We first obtain the set of all intersections of $R$ hyperplanes. For this we try to solve $\binom{N}{R}$ systems of $R$ equations. Let $\mathbf{M}_R \in \mathbb{R}^{R \times R}$ be the matrix obtained by choosing $R$ different rows $1, \ldots, R$ of $\mathbf{M} \in \mathbb{R}^{N \times R}$ (i.e., picking $R$ neurons), then we may find the corresponding intersection of $R$ hyperplanes by solving the following linear system of $R$ equations

$$\mathbf{z}_\cap = \mathbf{M}_R^{-1} \mathbf{h}_R \qquad \text{and} \qquad \mathbf{x}_\cap = \mathbf{Mz}_\cap.$$

which will always have a unique solution if all hyperplanes are in general position, as then all $\mathbf{M}_R$ have rank $R$. Each $\mathbf{x}_\cap$ has $2^R$ possible bordering linear regions. We can find the corresponding $\mathbf{D}_\Omega = \text{diag}([d_1, \ldots, d_N])$'s matrices of each of those subsections as follows. First $d_i = \mathbb{I}(\mathbf{x}_\cap < 0)$ for all $i <= N$. By construction $1, \ldots, R$ at $\mathbf{x}_\cap$ will be exactly at the threshold, by moving away from it $d_R$ can become either zero or one, depending on in which region we and up. Hence, the $2^R$ regions correspond to one in which either combination of neurons $1, \ldots R$ is active (meaning that it is above the threshold). We thus just have to check all combinations $d_1, \ldots, d_R \in \{0, 1\}^R$. Using this, we will find at most $\sum_{r=0}^R \binom{N}{r}$ unique configurations (as this is the maximal number of regions possible for $D = 1$). To find all the fixed points we hence have to solve Eq. 11 for each configuration. We thus end up with solving $\binom{N}{R}$ systems of $R$ linear equations to find all regions, and another $\sum_{r=0}^R \binom{N}{r} \in \mathcal{O}(N^R)$ (for fixed $R$) systems of $R$ linear equations to find all fixed points.

Let us now consider the case for $D > 1$. Note that an RNN with $N$ units and $D$ basis functions per unit, can be expanded to an RNN with $ND$ units with activation $\phi(\mathbf{x}_i) = \max(\mathbf{x}_i - \mathbf{h}_i, 0)$ ([24], Theorem 1). Any fixed point can then still be analytically computed using Eq. 11. We expand the network but keep track of all $\sum_r^R D^r \binom{N}{R}$ possible intersections. It still holds that from each

intersection, we can reach $2^R$ regions. In total, we will now find at most $\sum_{r=0}^{R} D^r \binom{N}{r}$ regions (Lemma 2). To find all the fixed points, we hence have to solve $\binom{N}{R} D^r + \sum_{r=0}^{R} \binom{N}{r} D^r$ systems of $R$ linear equations, which for constant $D$ and $R$ has a cost of $\mathcal{O}(N^R)$

Finally, let's consider the case when hyperplanes are not in general position (which is unlikely to happen when doing numerical optimization). If there are intersections of more than $R$ hyperplanes, we proceed as before, but in case the intersection of $R$ hyperplanes we are currently considering intersects additional hyperplanes, set the diagonal elements of $\mathbf{D}_\Omega$ corresponding to these additional hyperplanes arbitrarily to 1 (as intersections including the additional hyperplanes are considered separately). On the other hand, in case some hyperplanes are only part of intersections of less than $R$ hyperplanes (because they became parallel), we proceed as follows. Instead of considering only intersections of $R$ hyperplanes, we now also consider all possible intersections of $r$ hyperplanes, with $1 \leq r \leq R$. For this, we solve no more than $\sum_{r}^{R} \binom{N}{r}$ systems of $r$ equations. Let $\mathbf{M}_r \in \mathbb{R}^{r \times R}$ be the matrix obtained by choosing $r$ different linearly independent rows $1, \ldots, r$ of $\mathbf{M} \in \mathbb{R}^{N \times R}$; then we may find a point on the corresponding intersection of $r$ hyperplanes (note that the intersection itself can now also be a hyperplane) by to solving the following linear system of $r$ equations

$$\mathbf{z}_\cap = \mathbf{M}_r^\dagger \mathbf{h}_r \qquad \text{and} \qquad \mathbf{x}_\cap = \mathbf{M} \mathbf{z}_\cap.$$

with $\dagger$ being the pseudoinverse. We here now end up with solving no more than $\sum_{r}^{R} \binom{N}{r}$ systems of $r$ linear equations to find all regions, which has an equal cost in $N$ as the previous cases. $\qquad \square$

We here provide pseudocode. For simplicity, we restrict ourselved to the case of $D = 1$ and assume that the arrangement specified by $\mathbf{M}$ and $\mathbf{h}$ is in general position. This can be generalized to the general setting as presented in the proof.

---

**Algorithm 1:** Improved exhaustive search for all fixed points

**Data:** $\mathbf{N} \in \mathbb{R}^{N \times R}, \mathbf{M} \in \mathbb{R}^{N \times R}, \mathbf{h} \in \mathbb{R}^N$
**Result:** $z\_set$ set of all fixpoints, $D\_set$ the set of all relevant $\mathbf{D}_\Omega$ configurations.

$D\_set := \{\}$;
$z\_set := \{\}$;
$idx = [1, \ldots, N]$;

// *Find feasible configurations*
$idx\_comb$ = all $\binom{N}{R}$ combinations of indices $idx$;
**for** $(i_1, \ldots, i_R)$ *in* $idx\_comb$ **do**
$\quad$ $\mathbf{M}_R = \mathbf{M}[(i_1, \ldots, i_R), :]$;
$\quad$ $\mathbf{h}_R = \mathbf{h}[(i_1, \ldots, i_R)]$;
$\quad$ // $\mathbf{M}_R$ *is invertible as the arrangement is in general position*
$\quad$ $\mathbf{z}_\cap = \mathsf{solve}(\mathbf{M}_R, \mathbf{h}_R)$;
$\quad$ $\mathbf{x}_\cap = \mathbf{M}\mathbf{z}_\cap$;
$\quad$ $d\_init = \mathbf{x}_\cap > \mathbf{h}$;
$\quad$ **for** $(v_1, \ldots, v_R)$ *in* $\{0, 1\}^R$ **do**
$\quad\quad$ $d = d\_init[(i_1, \ldots, i_R)].set(v_1, \ldots, v_R)$ ;
$\quad\quad$ $\mathbf{D}_\Omega = \mathsf{diag}(d)$ ;
$\quad\quad$ $D\_set = D\_set \cup \{\mathbf{D}_\Omega\}$;
$\quad$ **end**
**end**

// *Find fixed points, for the at most* $\sum_{r=0}^{R} \binom{N}{r}$ *configurations*
**for** $\mathbf{D}_\Omega$ *in* $D\_set$ **do**
$\quad$ $\mathbf{z}^* = \mathsf{solve}(\mathbf{N}^\top \mathbf{D}_\Omega \mathbf{M} - \mathbf{I}, \mathbf{N}^\top \mathbf{D}_\Omega \mathbf{h})$ ;
$\quad$ $\mathbf{x}^* = \mathbf{M}\mathbf{z}^*$;
$\quad$ // *check if fixed point is consistent with assumed* $\mathbf{D}_\Omega$
$\quad$ **if** $\mathsf{diag}(\mathbf{x}^* > \mathbf{h}) == \mathbf{D}_\Omega$ **then**
$\quad\quad$ $z\_set = z\_set \cup \{\mathbf{z}^*\}$;
$\quad$ **end**
**end**

---

# B Additional figures & tables

## B.1 Additional statistics for Teacher-Student setups

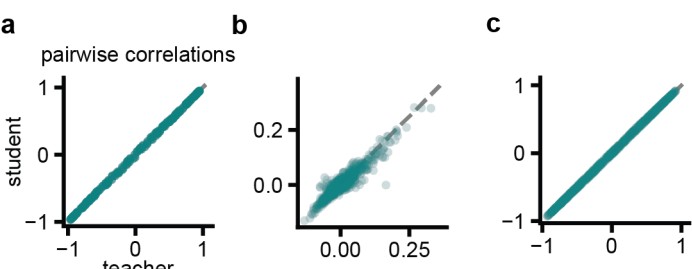

Supplementary Figure 4: **a-c)** Pairwise correlations between units of the modes for panel **a-c)** of Fig. 3, respectively. Note that **c** is computed over all conditions.

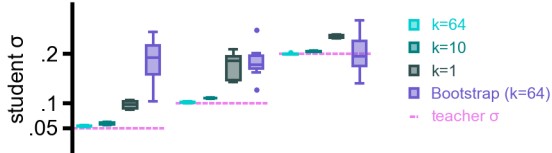

Supplementary Figure 5: Our method allows recovering the true latent noise in student-teacher setups. We repeated the experiment of Fig. 3**a** for teacher networks with three levels of latent noise, with diagonal covariances matrices $\Sigma_{\mathbf{z}} = \sigma^2 \mathbf{I}$. For each teacher we trained 5 student networks with varying number of particles (k), as well as using the bootstrap proposal (i.e., sampling from the prior). The standard deviations $\sigma$ of the latent noise process only matches between the student and teacher, if we use enough particles during training. The bootstrap proposal (with k=64) is not as reliable as the optimal proposal.

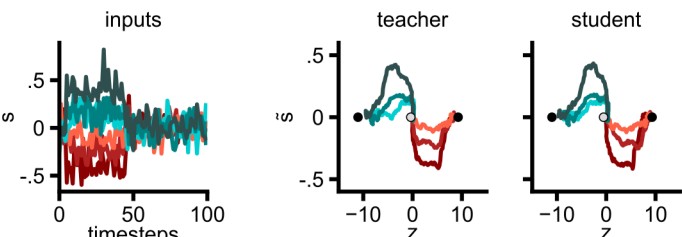

Supplementary Figure 6: To demonstrate that our method works with time-varying input, a rank-1 teacher RNN with 60 units was trained to report the sign of a time-varying stimulus (left). We then generated 400 trials of data with observation noise covariance $\Sigma_{\mathbf{x}} = .01\mathbf{I}$ and latent noise covariance $\Sigma_{\mathbf{x}} = .0025\mathbf{I}$. We trained a student on the observed activity of the teacher for 400 epochs. The matching latent dynamics of the student and teacher lie in the column space of the recurrent and input weights (right; coordinates $z$ and $\tilde{s}$, respectively; see Supplement C.2).

## B.2   EEG: Inferring full-rank RNNs

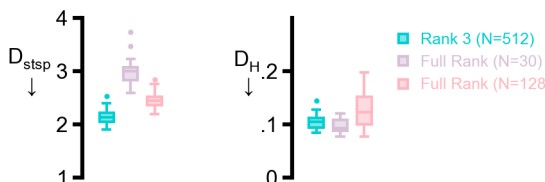

Supplementary Figure 7: We fit full-rank RNNs to EEG data, by parameterising the mean of the transition distribution as $F(\mathbf{z}_t) = a\mathbf{z}_t + (1-a)\mathbf{J}\phi(\mathbf{z}_t)$. We trained full-rank RNNs with 30 units (a roughly similar amount of parameters as our rank-3 RNNs with 512 units), as well as full-rank RNNs with 128 units (over 10 times more parameters). The KL divergence-based measure ($D_{\text{stsp}}$), between generated samples and data is worse for the full-rank RNNs, while also being less interpretable.

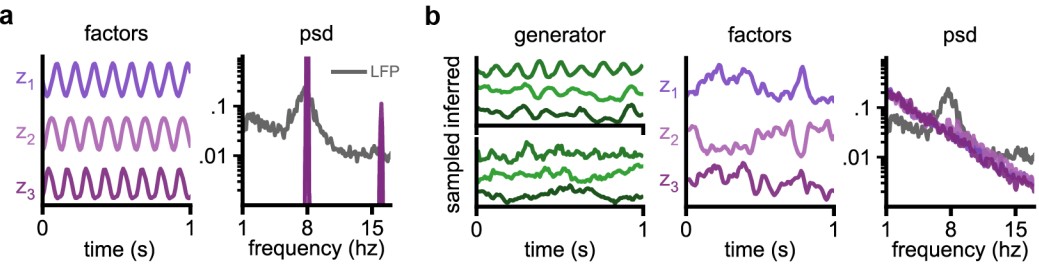

Supplementary Figure 8: Latent Factor Analysis via Dynamical Systems (LFADS) [7] is a current method that can be used to fit RNNs to neural data, and while it is generally used for inference, can also be used for generation. We here explored sampling from LFADS, both using the autonomous version, and when using the controller (i.e., stochastic inputs). We fitted LFADS to the HPC-2 dataset (using AutoLFADS for model selection [68]). **a**) In the case of an autonomous LFADS model, one samples an initial condition from the prior and then simulates a deterministic RNN forward. As on long sequences not all variability can be explained by variability in the initial condition, the latents end up not representing any variability that resembles the underlying system (cf. Fig. 5). **b**) In the case of a full LFADS model with the controller, one can sample both an initial condition and time-varying inputs from the controller's auto-regressive prior. Here the full model seemed to rely overly on the controller's data-inferred inputs for inference, which deviated quite strongly from the samples from the controller's auto-regressive prior. As a consequence, the generated latents do not seem to represent variability that is meaningful.

## B.3   HPC-2, additional results

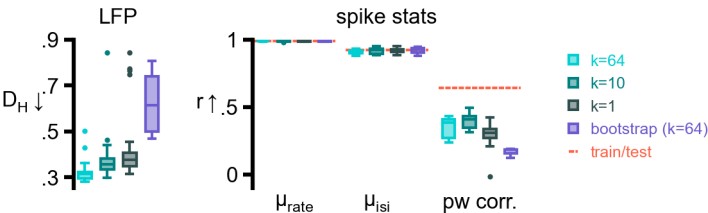

Supplementary Figure 9: The Hellinger distance ($D_H$) between the power spectrum of latents and LFP of HPC-2 is lower when using the bootstrap proposal or too few particles (left), and simulated data is slightly worse when using the bootstrap proposal (right)

## B.4 HPC-11, additional results

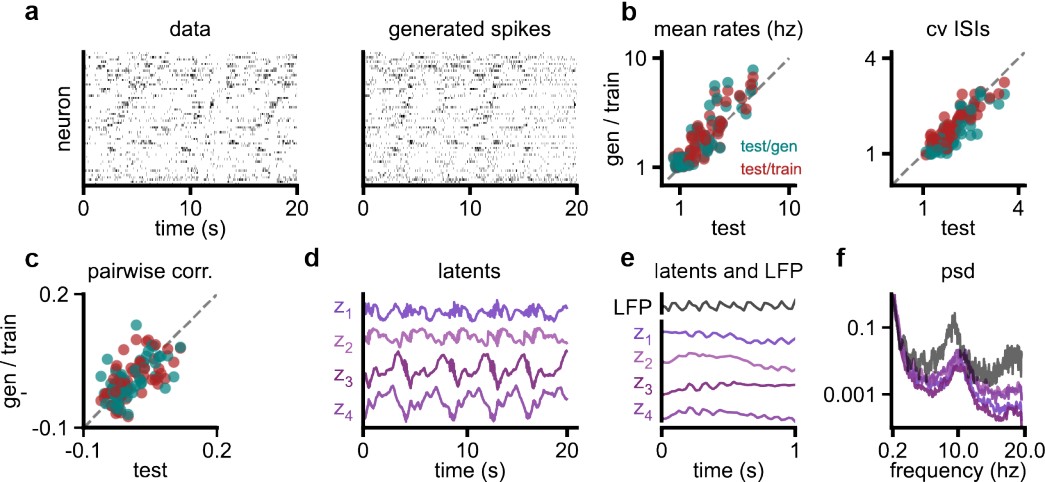

Supplementary Figure 10: **a**) We fit a rank-4 RNN to spikes recorded from rat hippocampus [50–52], and generate new samples from the RNN (right), taking only the part of the recording where the rat is running. **b**) Single neuron statistics. The mean rates and coefficient of variations of interspike interval (ISI) distributions of a long trajectory of data generated by the RNN (gen) match those of a held-out set of data (test). As a reference we additionally computed the same statistics between the train and test set. **c**) Population level statistics. The pairwise correlations between neurons for generated data and the test data. **d**) The corresponding latents generated by the RNN consists of 10Hz (fast theta) oscillations on top of slower oscillations. **e**) Latents with further zooming in (on time), shown together with the LFP signal. **f**) The power spectrum of latents sampled from the RNN, which show power at a slightly higher frequency than that of the LFP [54].

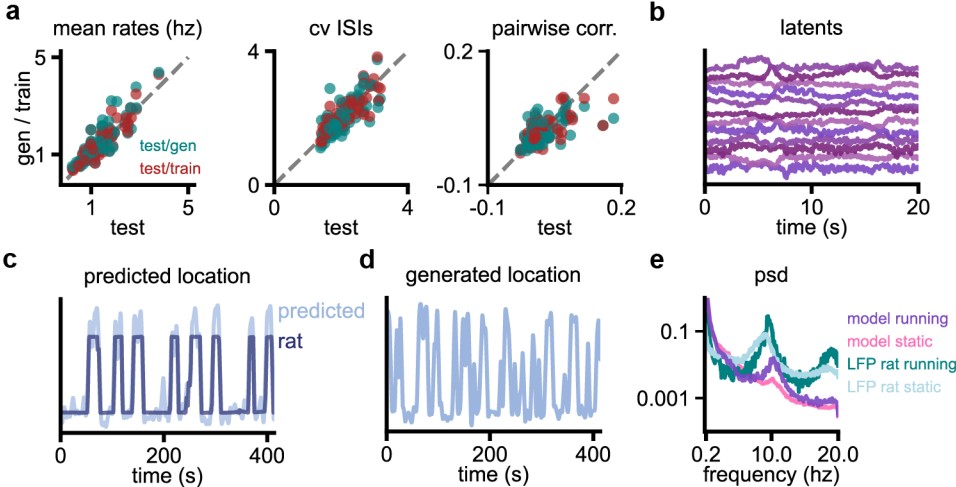

Supplementary Figure 11: We additionally fit a rank-12 RNN to a whole recording (2067 seconds resampled to 40 Hz) which includes long bouts where the rat is stationary. **a**) We generate a new sample from the RNN and obtain matching spike statistics. **b**) Generated latents by our model. **c**) Inferred posterior latents can again be used to predict the location of the rat on a held-out set. **d**) Using the same decoder on generated latents, we obtain a model that also predicts alternating bouts of stationarity and running. **e**) The mean power of the latents at theta frequency during running bouts is higher then during stationarity bouts. The increased theta power during running is also there in the LFP data — again with latent dynamics obtained from spiking data oscillating at a slightly higher frequency than the LFP.

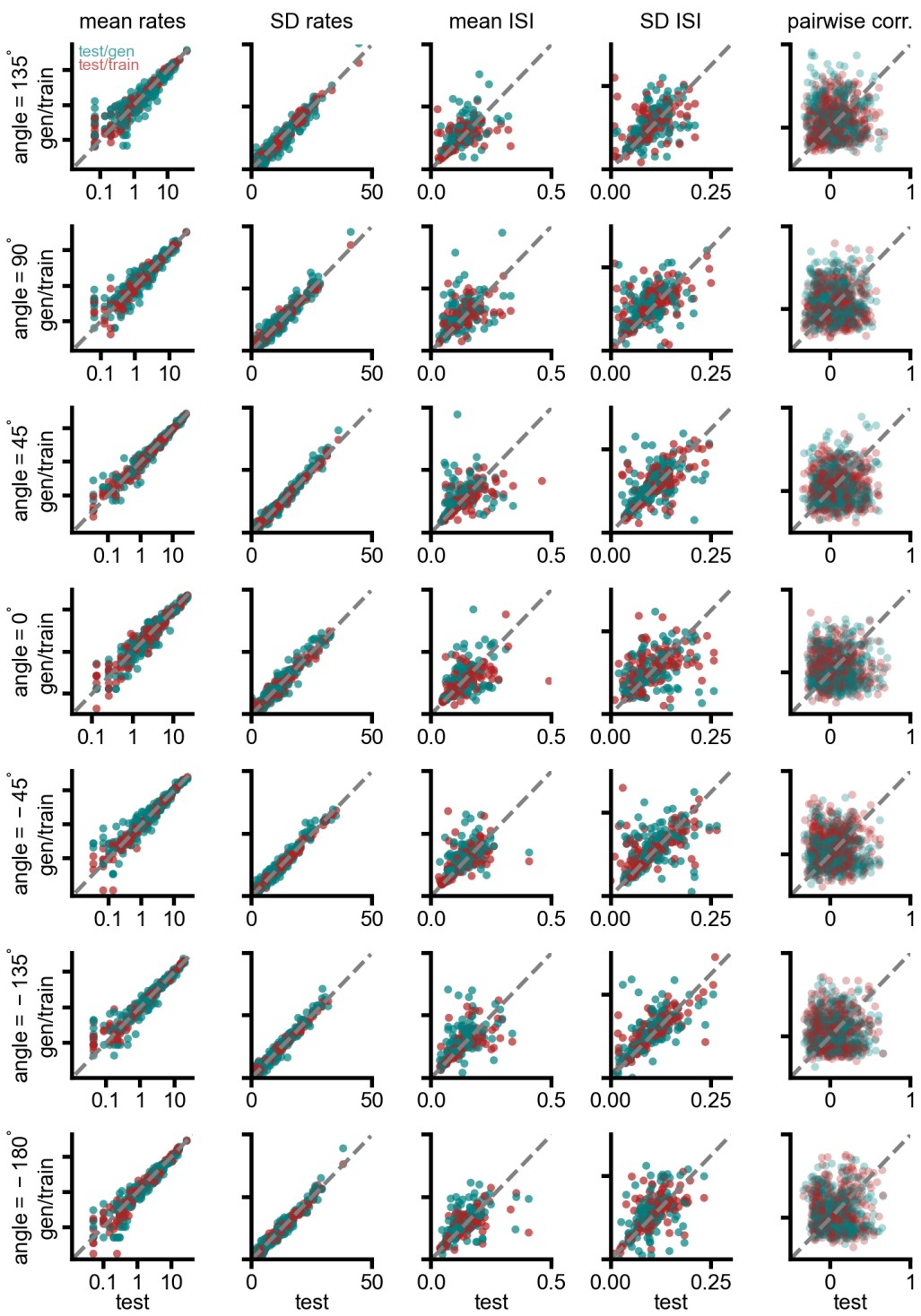

Supplementary Figure 12: Spiking statistics of model-generated (teal) and train data (brick red) compared against test data.

## B.5 Neural Latents Benchmark evaluation

We applied our method to the `MC_Maze` dataset of the Neural Latents Benchmark (NLB) [56] at 20 millisecond bin size (Table 2), by using our method to obtain expected Poisson rates, given a filtering posterior over latents. The benchmark evaluates methods on a number of metrics: 'co-bps' (co-smoothing bits-per-spike) assesses the quality of firing rate predictions for a set of held-out neurons that are unobserved in the test data, evaluated with the Poisson likelihood of the true spiking activity given the rate predictions. 'vel R2' evaluates how well the model's inferred firing rates can predict the subject's hand velocity. 'PSTH R2' evaluates how well peri-stimulus time histograms (PSTHs) computed from model-inferred rates match empirical PSTHs from the data. 'fp-bps' evaluates predictions on heldout timesteps (which we predict by running the RNN forward from the last data-inferred step). We found that our method outperforms classical methods (GPFA [69] and SLDS [30]) while certain state-of-the-art deep learning (LFADS [7, 68], Neural Data Transformer [70]) are slightly better than our method on the 'co-bps' metric, but our method matches them in the 'vel R2' metric (in case we include smoothing information in the proposal) . We do note that NLB metrics center around evaluating the quality of smooth rates *inferred* from spikes, which is not the central focus of our method, which is *generation*, i.e., sampling noisy trajectories that reproduce variability in the data. We here found that the quality of inference increased when using an non-causal CNN encoder as part of the proposal distribution, and additional gains might be obtained by also changing the target distribution to a smoothing (instead of a filtering) one [71].

While our method also has comparatively lower dimensionality than the other deep learning approaches, a latent dimensionality of 36 is still considerably higher than all networks considered in the Main text. We reason that we need a high number of latents, because the full `MC_Maze` dataset has a large number of conditions (108), spanning multiple maze-configurations, which may be difficult to fully model with autonomous low-dimensional latent dynamics.

Table 2: Performance of our method on the `MC_Maze` dataset of the Neural Latents Benchmark, 'dim' refers to the dimensionality of the model's underlying dynamics (where possible).

| method | dim | co-bps ↑ | vel R2 ↑ | PSTH R2 ↑ | fp-bps ↑ |
|---|---|---|---|---|---|
| Spike smoothing | 137 | 0.2076 | 0.6111 | −0.0005 | — |
| GPFA | 52 | 0.2463 | 0.6613 | 0.5574 | — |
| SLDS | 38 | 0.2117 | 0.7944 | 0.4709 | −0.1513 |
| LFADS | 100 | 0.3554 | 0.8906 | 0.6002 | 0.2454 |
| NDT | 274 | 0.3597 | 0.8897 | 0.6172 | 0.2442 |
| Ours | 36 | 0.3225 | 0.8479 | 0.5927 | 0.2184 |
| Ours (non-causal) | 36 | 0.3407 | 0.8902 | 0.5963 | 0.2417 |

## B.6 Stimulus-conditioning in monkey reaching task

For the experiment with stimulus-conditioned dynamics in the monkey reaching task, we tested the performance of models with and without the conditioning inputs. We found that the conditioning inputs allow the networks to perform better on velocity decoding at lower dimensionalities.

Table 3: Performance benefits of conditioning for monkey reaching task.

| conditioning | dim | vel R2 ↑ |
|---|---|---|
| w/o conditioning | 5 | $0.7897 \pm 0.0687$ |
| | 6 | $0.8944 \pm 0.0039$ |
| | 8 | $0.9085 \pm 0.0048$ |
| | 16 | $0.9196 \pm 0.0041$ |
| with conditioning | 5 | $0.8589 \pm 0.0493$ |
| | 6 | $0.9018 \pm 0.0114$ |

Following the analysis in Fig. 7, we also further visualized the model's match to the spiking statistics, including mean and standard deviation (SD) of spiking rate, and mean, SD, and coefficient of variation (CV) of inter-spike intervals. We observed a good match to the mean and SD of the spiking rate across all conditions. Match to ISI statistics is also quite reasonable given the noise observed between estimates of the statistics from train and test.

## B.7 Comparison to an approximate method for finding fixed points

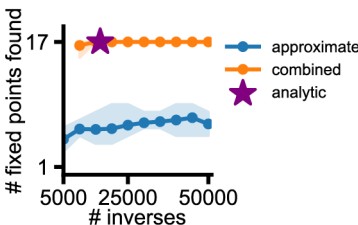

Supplementary Figure 13: Repetition of the experiment of Fig. 8, but now with a rank-2 RNN with 128 units. Again, we show the number of fixed points found as a function of the number of matrix inverses computed, with errorbars denoting the minimum and maximum amount of fixed points found over 20 independent runs of the algorithm.

Recently an approximate method for finding fixed points in piecewise-linear RNNs was proposed [25]. The method proceeds by randomly selecting a linear region (a configuration of $\mathbf{D}_\Omega$, see Supplement A.1.2) and calculating the corresponding fixed-point. If it is indeed a 'true' fixed point of the RNN (it is consistent with the assumed $\mathbf{D}_\Omega$), we store it. If the fixed point was inconsistent with the assumed $\mathbf{D}_\Omega$, we iteratively initialize $\mathbf{D}_\Omega$ according to the 'virtual' fixed point found and calculate the new fixed point corresponding to this $\mathbf{D}_\Omega$, until we either reach a 'true' fixed point or reach a certain amount of iterations. Then, we reinitialize at a randomly selected new configuration of $\mathbf{D}_\Omega$ and repeat the procedure.

Under some conditions, the approximate method can be shown to converge in linear time ($\|\mathbf{M}\tilde{\mathbf{N}}^\mathsf{T}\| + \|a\mathbf{I}\| \leq 1$) [8], where it will be faster than our exact method — however in general the convergence of the approximate method strongly depends on the dynamics of the networks. In particular, there are reasonable settings where the approximate method fails to find all fixed points, such as of a rank-2, 128 unit RNN with 17 fixed points (trained similarly to the teacher RNN of Fig. 3**c**; Fig. S13). While an in-depth study of the approximate method is out of scope, we hypothesize that the failure to converge is because when initializing with randomly selected $\mathbf{D}_\Omega$s out of $(D+1)^N$ possible configurations, the approximate method tends to converges to the same set of $\mathbf{D}_\Omega$s.

Our method is completely independent of the dynamics of the system and has a fixed cost, after which one is guaranteed that all fixed points are found. However, we do note that there can be scenarios where our exact method is still too costly. In this scenario, we propose to use the approximate method, with one adjustment - we first pre-compute the subset of $\sum_r^R D^r \binom{N}{r}$ configurations that can contain fixed points, and then initialize the approximate method using randomly selected $\mathbf{D}_\Omega$ from this subset. Empirically, this leads to better convergence in at least some scenarios (Fig. 8, Fig. S13)

For the approximate method, we used code from https://github.com/DurstewitzLab/CNS-2023, which was released with the GNU General Public License.

# C Additional details of low-rank RNNs

## C.1 Discretisation

Given

$$\tau \frac{d\mathbf{z}}{dt} = -\mathbf{z}(t) + \mathbf{N}^\mathsf{T}\phi(\mathbf{M}\mathbf{z}(t)) + \Gamma_\mathbf{z}\xi(t),$$

Using the Euler–Maruyama method with timestep $\Delta_t$:

$$\mathbf{z}_{t+1} = (1 - \frac{\Delta_t}{\tau})\mathbf{z}_t + \frac{\Delta_t}{\tau}\mathbf{N}^\mathsf{T}\phi(\mathbf{Mz}_t) + \frac{\sqrt{\Delta_t}}{\tau}\Gamma_\mathbf{z}\epsilon_t,$$

and with $\epsilon_t \sim \mathcal{N}(0, \mathbf{I})$, define $a = 1 - \frac{\Delta_t}{\tau}$, $\tilde{\mathbf{N}} = \frac{\Delta_t}{\tau}\mathbf{N}$, and $\Sigma_\mathbf{z} = \frac{\Delta_t}{\tau^2}\Gamma_\mathbf{z}\Gamma_\mathbf{z}^\mathsf{T}$, we obtain the transition distribution used in our experiments. Note the slight 'overloading' of $t$ here, as the discrete time indice $t$ of e.g., $\mathbf{z}_t$ corresponds to the continuous time $\mathbf{z}((t-1)\Delta_t)$.

## C.2    Conditional generation

Given input weights $\mathbf{H} \in \mathbb{R}^{N \times N_s}$ and stimulus $\mathbf{s} \in \mathbb{R}^{N_s}$, we define our model as

$$\tau\frac{d\mathbf{x}}{dt} = -\mathbf{x}(t) + \mathbf{J}\phi(\mathbf{x}(t)) + \mathbf{Hs}(t) + \xi_\mathbf{x}.$$

Using the same assumptions as before, $\mathbf{x}$ can be described by $R + N_s$ variables

$$\tau\frac{d\mathbf{z}}{dt} = -\mathbf{z}(t) + \mathbf{N}^\mathsf{T}\phi(\mathbf{Mz}(t) + \mathbf{H}\tilde{\mathbf{s}}(t)) + \xi_\mathbf{z},$$

$$\tau\frac{d\tilde{\mathbf{s}}}{dt} = -\tilde{\mathbf{s}}(t) + \mathbf{s}(t),$$

with $\mathbf{x} = \mathbf{Mz} + \mathbf{H}\tilde{\mathbf{s}}$, and $\begin{bmatrix}\mathbf{z}\\\tilde{\mathbf{s}}\end{bmatrix} = ([\mathbf{M}, \mathbf{H}]^\mathsf{T}[\mathbf{M}, \mathbf{H}])^{-1}[\mathbf{M}, \mathbf{H}]^\mathsf{T}\mathbf{x}$.

We can write the distribution generated after discretization as:

$$p(\mathbf{z}_{1:T}, \mathbf{y}_{1:T}, \tilde{\mathbf{s}}_{1:T} | \mathbf{s}_{1:T-1}) = p(\mathbf{z}_1)p(\tilde{\mathbf{s}}_1)\prod_{t=2}^T p(\tilde{\mathbf{s}}_t \mid \mathbf{s}_{t-1})p(\mathbf{z}_t \mid \tilde{\mathbf{s}}_{t-1}, \mathbf{z}_{t-1})\prod_{t=1}^T p(\mathbf{y}_t \mid \tilde{\mathbf{s}}_t, \mathbf{z}_t),$$

$$p(\mathbf{z}_t \mid \tilde{\mathbf{s}}_{t-1}, \mathbf{z}_{t-1}) = \mathcal{N}(F(\tilde{\mathbf{s}}_{t-1}, \mathbf{z}_{t-1}), \Sigma_\mathbf{z}), \quad p(\mathbf{z}_1) = \mathcal{N}(\mu_{\mathbf{z}_1}, \Sigma_{\mathbf{z}_1}),$$

$$p(\tilde{\mathbf{s}}_t \mid \tilde{\mathbf{s}}_{t-1}, \mathbf{s}_{t-1}) = \delta(a\tilde{\mathbf{s}}_{t-1} + (1-a)\mathbf{s}_{t-1}), \ p(\tilde{\mathbf{s}}_1) = \delta(\mathbf{0}),$$

where the mean of the transition distribution is $F(\tilde{\mathbf{s}}_t, \mathbf{z}_t) = a\mathbf{z}_t + \tilde{\mathbf{N}}^\mathsf{T}\phi(\mathbf{Mz}_t + \mathbf{H}\tilde{\mathbf{s}}_t)$.

For constant input $\mathbf{s}$, $\tilde{\mathbf{s}}$ will converge to $\mathbf{s}$, and we can ignore the additional $N_s$ variables, assuming $\mathbf{x}(0) = \mathbf{Mz}(0) + \mathbf{s}$. Similarly if $\mathbf{s}$ varies on a time scale slower than $\tau$, $\mathbf{s} \approx \tilde{\mathbf{s}}$ is a good approximation [21]. Here, for experiments where the input is a constant context signal (Fig. 7), we substitute $\mathbf{s}$ for $\tilde{\mathbf{s}}$ and consider the $R$ dimensional system described by $\mathbf{z}$ (which now has additional conditioning on $\mathbf{s}$):

$$p(\mathbf{z}_{1:T}, \mathbf{y}_{1:T} \mid \mathbf{s}_{1:T}) = p(\mathbf{z}_1)\prod_{t=2}^T p(\mathbf{z}_t \mid \mathbf{s}_{t-1}, \mathbf{z}_{t-1})\prod_{t=1}^T p(\mathbf{y}_t \mid \mathbf{s}_t, \mathbf{z}_t),$$

$$p(\mathbf{z}_t \mid \mathbf{s}_{t-1}, \mathbf{z}_{t-1}) = \mathcal{N}(F(\mathbf{s}_{t-1}, \mathbf{z}_{t-1}), \Sigma_\mathbf{z}), \ p(\mathbf{z}_1) = \mathcal{N}(\mu_{\mathbf{z}_1}, \Sigma_{\mathbf{z}_1}),$$

where $F(\mathbf{s}_t, \mathbf{z}_t) = a\mathbf{z}_t + \tilde{\mathbf{N}}^\mathsf{T}\phi(\mathbf{Mz}_t + \mathbf{Hs}_t)$.

## C.3    Linear transformations of the latent space and orthogonalisation

Given

$$\mathbf{x}_{t+1} = a\mathbf{x}_t + \mathbf{M}\tilde{\mathbf{N}}^\mathsf{T}\phi(\mathbf{x}_t) + \epsilon_\mathbf{x}$$

$$\mathbf{z}_{t+1} = a\mathbf{z}_t + \tilde{\mathbf{N}}^\mathsf{T}\phi(\mathbf{Mz}_t) + \epsilon_\mathbf{z}$$

with $\epsilon_\mathbf{z} \sim \mathcal{N}(0, \Sigma_\mathbf{z})$, $\epsilon_\mathbf{x} \sim \mathcal{N}(0, \mathbf{M}\Sigma_\mathbf{z}\mathbf{M}^\mathsf{T})$. We can do any linear transformation of the latent dynamics $\mathbf{z}$: $\hat{\mathbf{z}} = \mathbf{Az}$, as long as $\mathbf{A}$ has rank $R$, without changing the neuron activity $\mathbf{x}$. To see this, define $\hat{\mathbf{M}} = \mathbf{MA}^{-1}$, $\hat{\mathbf{N}} = \mathbf{A}\tilde{\mathbf{N}}$, and $\epsilon_{\hat{\mathbf{z}}} \sim \mathcal{N}(0, \mathbf{A}\Sigma_\mathbf{z}\mathbf{A}^T)$, giving us:

$$\mathbf{x}_{t+1} = a\mathbf{x}_t + \hat{\mathbf{M}}\hat{\mathbf{N}}^\mathsf{T}\phi(\mathbf{x}_t) + \epsilon_\mathbf{x}$$

$$\hat{\mathbf{z}}_{t+1} = a\hat{\mathbf{z}}_t + \hat{\mathbf{N}}^\mathsf{T}\phi(\hat{\mathbf{M}}\hat{\mathbf{z}}_t) + \epsilon_{\hat{\mathbf{z}}},$$

which will leave $\mathbf{x}$ unchanged, while our latents $\mathbf{z}$ are expressed in a new basis. We typically got a more interpretable visualization of the latents by orthonormalising the columns of $\mathbf{M}$. Thus we applied for all visualisations after training $\mathbf{A} = \mathbf{U}^\mathsf{T}\mathbf{M}$, with $\hat{\mathbf{M}} = \mathbf{U}$, where $\mathbf{U}$ are the first $R$ left singular vectors of $\mathbf{J} = \mathbf{MN}^\mathsf{T}$.

# D Details of empirical experiments

## D.1 Training details

### D.1.1 Initialisation

Our models are (unless noted otherwise) initialized as follows:

$$
\begin{aligned}
\tilde{\mathbf{N}}_{ij} &\sim \mathcal{U}_{[-\frac{1}{\sqrt{N}}, \frac{1}{\sqrt{N}}]}, \\
\mathbf{M}_{ij} &\sim \mathcal{U}_{[-\frac{1}{\sqrt{R}}, \frac{1}{\sqrt{R}}]}, \\
\mathbf{W}_{ij} &\sim \mathcal{N}(0, \frac{2}{R}), \\
\mathbf{H}_{ij} &\sim \mathcal{U}_{[-\frac{1}{\sqrt{N_{inp}}}, \frac{1}{\sqrt{N_{inp}}}]}, \\
\mathbf{h}_i &\sim \mathcal{U}_{[-\frac{1}{\sqrt{N}}, \frac{1}{\sqrt{N}}]}, \\
\mathbf{b} &\leftarrow \mathbf{0}, \\
a &\leftarrow .9, \\
\Sigma_{\mathbf{z}} &\leftarrow .01\mathbf{I}, \\
\Sigma_{\mathbf{z}_1} &\leftarrow \mathbf{I}, \\
\mu_{\mathbf{z}_1} &\leftarrow \mathbf{0},
\end{aligned}
$$

where $\mathbf{W}$ and $\mathbf{b}$ are the output weights and biases respectively. For Gaussian observations we initialise $\Sigma_{\mathbf{y}} \leftarrow .01\mathbf{I}$.

For experiments with Poisson observations, we jointly optimized a (usually causal) CNN encoder as part of the proposal distribution. The CNN was conditioned on observations and predicted the mean and $\log$ variance of a normal distribution. It consisted of common initial layers consisting of 1D convolutions, with a GeLU activation function, and a separate output convolution for the predicted mean and (log) variance. The CNN was initialized to the Pytorch [72] defaults, except for the bias of the $\log$ variance output layer, to which we added a $\log(.01)$ term, such that the output matches the initially predicted variance of the RNN. The exact number of layers and channels are reported in the sections for each experiment.

For the teacher-student setups, we used as non-linearity $\phi(\mathbf{x}_i) = \max(\mathbf{x}_i - \mathbf{h}_i, 0)$ for both the students and the teachers, and for all experiments with real-world data, we used the 'clipped' $\phi(\mathbf{x}_i) = \max(\mathbf{x}_i + \mathbf{h}_i, 0) - \max(\mathbf{x}_i, 0)$ [8].

### D.1.2 Parameterisation

We constrain $a$ to be between $0$ and $1$ by instead optimising $\tilde{a}$ with the following (sigmoidal) parameterisation $a = \exp(-\exp(\tilde{a}))$ [73]. In experiments with the optimal proposal, we estimate the full $\Sigma_{\mathbf{z}}$, which we constrain to be symmetric positive definite, by optimizing a lower triangular matrix $\mathbf{C}$ such that $\Sigma_{\mathbf{z}} = \mathbf{C}\mathbf{C}^T$, where we additionally constrain the diagonal of $\mathbf{C}$ to be positive using $\mathbf{C}_{ii} = \exp(\tilde{\mathbf{C}}_{ii}/2)$. For all diagonal covariances, we parameterise the diagonal elements using $\Sigma_{ii} = \exp(\tilde{\Sigma}_{ii})$. For Poisson observations, we apply a Softplus function to rectify the predicted rate.

### D.1.3 Optimisation

During training we minimise the variational SMC ELBO [33–35] (Eq.7) with stochastic gradient descent, using the RAdam [74] optimiser in Pytorch [72]. We generally use an exponentially decaying learning rate (details under each experiment).

## D.2 Teacher student experiments

### D.2.1 Dataset description

We created datasets by first training 'teacher' RNNs to perform a task and then generating observations by simulating the trained teacher RNNs.

For Fig. 3**a**, **b** we used code from [22] (https://github.com/mackelab/phase-limit-cycle-RNNs, Apache licence) to train rank-2 RNNs to produce oscillations, using a sine-wave with a periodicity of 50 time-steps as a target and an additional L2 regularisation on the rates. After training, we extracted the recurrent weights $\mathbf{M}$, $\mathbf{N}$ and biases $\mathbf{h}$, orthonormalized the columns of $\mathbf{M}$, and created a dataset by simulating the model for 75 timesteps, with $\Sigma_{\mathbf{z}} = .04\mathbf{I}$. For Fig. 3**a** we used $N = 20$ units and generated observations according to $G = \mathcal{N}(\mathbf{Mz}_t, \Sigma_{\mathbf{y}})$, with $\Sigma_{\mathbf{y}} = .01\mathbf{I}$. Fig. 3**b** we used $N = 40$ units and generated observations according to $G = \mathsf{Pois}(\mathsf{Softplus}(w\mathbf{Mz}_t - b))$, with $w = 4$ and $b = 3$.

For Fig. 3**c**, we followed a similar procedure but now trained the teacher RNN on a task where it has to use input. After an initial period of 25 time steps, a stimulus was presented for 25 timesteps consisting of $[\sin(\theta), \cos(\theta)]^{\mathsf{T}}$, where $\theta$ was randomly selected every trial out of 8 fixed angles. The RNN was tasked to produce output that equals the transient stimulus for the next 100 time-steps. Here we used $N = 60$ units, $\Sigma_{\mathbf{z}} = .0025\mathbf{I}$ and generated observations according to $G = \mathcal{N}(\mathbf{Mz}_t, \Sigma_{\mathbf{y}})$, with $\Sigma_{\mathbf{y}} = .0025\mathbf{I}$. The training data for the student RNN was included for each trial the corresponding stimulus.

### D.2.2 Training details

The 'student' RNNs had $20, 40, 60$ units, respectively and rank $R = 2$, matching that of the teacher RNNs. For Fig. 3**a**, **c**. The observation model was a linear Gaussian according to $G = \mathcal{N}(\mathbf{Mz}_t, \Sigma_{\mathbf{y}})$, and we used the optimal proposal distribution. For Fig. 3**b** we used $G = \mathsf{Pois}(\mathsf{Softplus}(\mathbf{WMz}_t - \mathbf{b}))$, with $\mathbf{W}$ a diagonal matrix (scaling the output of each unit individually). For Fig. 3**b**, we used a causal CNN encoder as part of the proposal distribution. It consisted of 3 layers, with kernel sizes $(21, 11, 1)$, and channels $(64, 64, 2)$. We used (causal) circular padding.

For all three experiments, we used $k = 64$ particles, batch-sizes of 10, and decreased the learning rate exponentially from $10^{-3}$ to $10^{-5}$. For Fig. 3**a** we trained for 1000 epochs of 200 trials, for Fig. 3**b** for 1500 epochs of 400 trials and for Fig. 3**c** for 200 epochs of 800 trials. We used a workstation with a NVIDIA GeForce RTX 3090 GPU for these runs. One model took about 3 to 4 hours to finish training.

### D.2.3 Evaluation setup

For Fig. 3 we generated long trajectories of $T = 10000$ time-steps of data for both the student and teacher RNNs. To facilitate visual comparisons between student and teacher dynamics, we also orthonormalized the columns of the students weights $\mathbf{M}$ after training, and for Fig. 3**a**,**c** picked signs of the columns of $\mathbf{M}$ such that the student and teacher match (note that after orthormalizing, the columns of $\mathbf{M}$ are equal to the non-zero singular vectors of the full weight matrix $\mathbf{J}$, which are only unique up to a sign flip). As noted before, this leaves the output of the model unchanged. The autocorrelation in Fig. 3**a** was computed by convolving a sequence of lag$= 120$ steps of data with itself (with duration $2 \times$lag), and normalising such that lag$=0$ corresponds to a correlation of 1. We repeated this for 80 sequences starting at different time-points of the whole trajectory.

### D.3 EEG data

### D.3.1 Dataset description

We used openly accessible electroencephalogram (EEG) data from [42, 43] ( https://www.physionet.org/content/eegmmidb/1.0.0/, ODC-BY licence). The data was recorded from a human subject sitting still with eyes open (session S001R01), and was sampled at 160 Hz. Like [8], we used the full 1 minute of recording, but unlike [8], we did not smooth the data (but just standardized the data). Thus, to compare our performance to [8], who ran their evaluation using the smoothed data, we smoothed our generated samples equivalently, using a Hann filter with a window length of 15-time bins, so that we can also compare our samples to the smoothed data.

### D.3.2 Training details

We used $N = 512$ units, and rank $R = 3$. The observation model was a linear Gaussian conditioned on the hidden state and we used the optimal proposal distribution. We trained for 1000 epochs consisting of 50 batches of size of 10, and $k = 10$ particles. The learning rate was decreased

exponentially from $10^{-3}$ to $10^{-6}$. Models were trained using NVIDIA RTX 2080 TI GPUs on a compute cluster. A single model took between 4 and 5 hours to finish training.

### D.3.3   Evaluation setup

We used our RNN to generate one long trajectory of $T = 9760$ steps of data, $\mathbf{y}_t$ (after discarding the first 2440 steps), which we compare to the EEG data, $\hat{\mathbf{y}}_t$, using two evaluation measures from [8, 24] (using code from https://github.com/DurstewitzLab/GTF-shPLRNN, GNU General Public License):

$\mathbf{D}_{\mathsf{stsp}}$: This is an estimate of the KL divergence between the ground truth and generated states. To compute this, we obtained kernel density estimates of the probability density functions (over states, not time), using a Gaussian kernel with standard deviation $\sigma = 1$. We get for the EEG data: $\hat{p}(\mathbf{y}) = \frac{1}{T}\sum_{t=1}^{T}\mathcal{N}(\hat{\mathbf{y}}_t, \mathbf{I})$, and for the generated data $\hat{q}(\mathbf{y}) = \frac{1}{T}\sum_{t=1}^{T}\mathcal{N}(\mathbf{y}_t, \mathbf{I})$. We then used the following Monte Carlo estimate of the KL divergence: $D_{\mathsf{stsp}} \approx \frac{1}{n}\sum\log\frac{\hat{p}(\hat{\mathbf{y}}^i)}{\hat{q}(\hat{\mathbf{y}}^i)}$, using $n = 1000$ samples $\hat{\mathbf{y}}^i$ drawn randomly from the EEG data.

$\mathbf{D}_{\mathbf{H}}$: This is an estimate of the difference in power spectra between the ground truth and generated states. We first computed for each data dimension the spectra $\hat{\mathbf{y}}_\omega^i, \mathbf{y}_\omega^i$ for the EEG and generated data, respectively. We used a Fast Fourier Transform, smoothed the estimates with a Gaussian kernel with standard deviation $\sigma = 20$, and normalized the spectra so they sum to 1. We computed the mean of the Hellinger distances between the spectra: $D_H = \frac{1}{64}\sum_i^{64}\frac{1}{\sqrt{2}}\|\sqrt{\hat{\mathbf{y}}_\omega^i} - \sqrt{\mathbf{y}_\omega^i}\|$.

## D.4   Hippocampus HC-2

### D.4.1   Dataset description

We used openly accessible neurophysiological data recorded from layer CA1 of the right dorsal hippocampus [47, 48] (https://crcns.org/data-sets/hc/hc-2/about-hc-2. Signals were recorded as the rats engaged in an open field task, chasing drops of water or pieces of food that were randomly placed. We used the session ec013.527 from rat ID ec13, which is approximately 1062 seconds long. From 37 units (neurons) we used 21 neurons that have maximal spike counts, discarding the rest of the comparatively silent neurons. We binned the spike data to 10ms. We used the first 80 percent of the data for training, and the rest was saved for testing purposes.

### D.4.2   Training details

We used $N = 512$ units, and rank $R = 3$ for the run that was used in our Fig. 5. We used a causal CNN encoder as part of the proposal distribution, which consisted of 3 layers with kernel sizes (150, 11, 1), with (64, 64, 3) channels. During our study, we swept over multiple ranks and found that theta oscillations consistently emerged from rank 3 onwards, after which reconstruction accuracy was relatively stable. For each rank, we used three different seeds and two different first layer sizes for the encoder, 25 or 150. The duration of a randomly sampled trial (sequence length) from the whole data was 94 time steps when the first layer size was 25, and 219 when the first layer size was 150. We, however, also found that the choice of the duration did not affect the results much. We trained the model using 3000 epochs, each epoch consisting of 3000 trials with 64 batches and $k = 64$ particles. The learning rate was decreased exponentially from $10^{-3}$ to $10^{-6}$. A single model took approximately 21 hours to finish training on a NVIDIA RTX 2080 TI GPU on a compute cluster.

### D.4.3   Evaluation setup

We used our RNN to generate data that matches the duration of the test data, which is 20810 time steps ($\sim$208 s) (after discarding the first 1000 steps). We compare different spike statistics of generated data with test data, and for comparison purposes, we also compared the same statistics measurements between train and test data as well. We calculated the mean firing rate of each neuron, mean of ISI distributions, and pairwise correlations. We used a band-pass filter 1-40 Hz for the latents and the LFP signal before calculating the powerspectrogram (Fig.5**e**).

### D.5 Hippocampus HC-11

#### D.5.1 Dataset description

We used openly accessible neurophysiological data recorded from hippocampal CA1 region [50–52] (https://crcns.org/data-sets/hc/hc-11/about-hc-11). We used the subset of the dataset called the *maze* epoch, where a rat was running on a 1.6-meter linear track, with rewards located at each end (left and right). Throughout this task, neural activity was recorded from 120 identified pyramidal neurons. As in [13], we only used 60 neurons that had sufficient activity and discarded rest of the units. We used code from [53] (https://github.com/zhd96/pi-vae) to preprocess the spike data, and only use data corresponding to the rat running and the location data being available for the results shown in Fig. 6 and Fig. S10. For the model shown in Fig. S11 we used the last 1350s of data of the Maze epoch, which also includes bouts where the rat is stationary. We used 25ms bins.

#### D.5.2 Training details

We used $N = 512$ units, and rank $R = 4$ for Fig. 6 and Fig. S10 and rank $R = 12$ for Fig. S11. We used a causal CNN encoder with zero padding with 3 layers (24, 11, 1), (64, 64, 4) channels, and 3 layers (150, 11, 1), (128, 64, 12) channels, respectively. The models were trained for 3000 epochs, each epoch having 3000 trials with a sequence length of 94 time bins (2.35 s) and 219 bins (5.48 s), respectively. We used batch sizes of 64 and $k = 64$ particles. The learning rate was decreased exponentially from $10^{-3}$ to $10^{-6}$. A single model took approximately 21 hours to finish training on a NVIDIA RTX 2080 TI GPU on a compute cluster.

#### D.5.3 Evaluation setup

We used our RNN to generate data that matches the duration of the test data, which is 4289 time steps ($\sim$107 s) (after discarding the first 1000 steps) for Fig. 6 and Fig. S10 and 16539 time steps ($\sim$413 s) for Fig. S11. We calculated the mean firing rate of each neuron, coefficient of variations of ISI distributions, and pairwise correlations. For the $R^2$ reported in the Main text, we fit a ridge regression model to the posterior latent variables on the training data, after smoothing with a Hann window of size 100, and apply the regression model to latents inferred from the test data.

### D.6 Monkey Reach

#### D.6.1 Dataset description

We used the publicly available `MC_Maze` dataset from the Neural Latents Benchmark (NLB) [56] (https://dandiarchive.org/dandiset/000128, CC-BY-4.0 licence). The data were recorded from a macaque performing a delayed center-out reaching task with barriers, resulting in a variety of straight and curved reaches. For simplicity, we took only the trials with no barriers and thus straight reach trajectories, resulting in 592 training trials and 197 test trials. We binned the data at 20 ms and aligned each trial from 250 ms before to 450 ms after movement onset.

To create conditioning inputs for the model, we took the x and y coordinates of the target position for each trial and scaled them to be between $-1$ and $1$. We then provide this scaled target position as constant context input to the RNN for the duration of the trial.

#### D.6.2 Training details

We ran a random search of 30 different models with rank $r \in 3, 4, 5, 6$ and particle number $k \in 16, 32, 64$. All models had 512 units and used a causal CNN encoder with kernel sizes $(14, 4, 2)$ and channels $(128, 64, r)$. We used (causal) reflect padding. We trained each model for up to 2000 epochs, terminating training early if no improvement was seen for 50 epochs. Each model took around 3 to 4 hours to train on an NVIDIA RTX 2080 TI GPU on a compute cluster. Seeing that a rank of 5 was sufficient for velocity decoding $R^2 \approx 0.9$, we took the best-performing rank-5 model for subsequent analyses.

### D.6.3 Evaluation setup

For qualitative evaluation of replication of cross-condition differences, we grouped the reach targets in the data into 7 conditions, one at each corner and the midpoint of each edge of the rectangular reach plane, excluding the midpoint directly at the bottom. We then generated data from the model RNN using conditioning inputs from the test trials of the real data. Then, for the test data and the model-generated data, we computed mean firing rate and inter-spike interval for each neuron for each condition. We then computed correlation distance ($1 - r$, where $r$ is the Pearson correlation coefficient) on the neuron statistics between conditions in the test data and model-generated data.

For generation of data for Fig. 7**d**, **e**, we selected target locations by choosing angles from 0 to 360°, evenly spaced by 22.5°, and determined the corresponding reach endpoint on a square spanning from $(-1, -1)$ to $(1, 1)$. We then constructed conditioning inputs similar to the real data using these target locations and simulated the RNN with them. To decode the reaches, we used a linear decoder trained from inferred firing rates to reach velocity from the real data.

### D.7 Neural Latents Benchmark

### D.7.1 Dataset description

We again used the publicly available `MC_Maze` dataset from NLB (see Supplement D.6.1). We resampled the data to 20 ms bin size and followed the standard data preprocessing procedures for the benchmark, as described in [56].

### D.7.2 Training details

We ran a random search of 30 different models with varying rank from 12 to 40 and particle number $k \in 16, 32, 64$. All models had 512 units and used a used a CNN encoder with kernel sizes $(14, 4, 2)$ and channels $(128, 64, 36)$, either with causal reflect padding or acausal zero padding. We trained each model for up to 2000 epochs, terminating training early if no improvement was seen for 50 epochs. Each model took around 10 to 12 hours to train on an NVIDIA RTX 2080 TI GPU on a compute cluster.

Because the primary task of the benchmark is co-smoothing, i.e., prediction of held-out neuron firing rates from held-in neurons, we provide the encoder with only the activity of held-in neurons. However, the observation likelihood component of the ELBO is computed on all neurons, held-in and held-out.

After training, we selected the model with the best co-smoothing score on the validation split and submitted its predictions to the benchmark for the final evaluation.

### D.7.3 Evaluation setup

Automated evaluation was performed on the benchmark platform, as described in [56].

We used for the prediction at timestep $t$, the expected Poisson rate of held-out neurons, conditioned on the activity of held-in neurons at the current and previous timesteps, by making use of the filtering Posterior (Eq. 3). We averaged over 32 sets of trajectories with 192 particles each.

