# OpenReview forum: "Inferring stochastic low-rank recurrent neural networks from neural data"
_NeurIPS.cc/2024/Conference — NeurIPS 2024 poster_

### Official Review · Reviewer_Strk · 2024-07-03

**Soundness:** 4
**Presentation:** 3
**Contribution:** 3
**Rating:** 6
**Confidence:** 2

**Summary:**

The authors propose fitting stochastic low-rank RNNs to neural recordings using variational sequential Monte Carlo methods. Such techniques permit modeling of noisy sequences (i.e., trial-to-trial variability), identification of a low-dimensional latent dynamical system, generative sampling of neural trajectories with realistic variability, and interpretation via fixed point analysis. The technique is applied successfully to recover the ground truth dynamics in two synthetic systems, and then to model EEG recordings, hippocampal spiking data, and motor cortical spiking data.

**Strengths:**

- Originality: The technique presented appears original in its combination of existing ideas from low-rank RNNs, variational inference, and sequential Monte Carlo.
- Quality: Thoughtful comparisons were made to existing approaches. The results improve upon state-of-the-art techniques (e.g., Generalised Teacher Forcing) in certain settings.
- Clarity: The figures and tables are clearly presented and quite interpretable. Much of the writing is clear, although see Weaknesses and Questions for suggestions here.

**Weaknesses:**

- Section 2.2 could benefit from being made more accessible to readers who are not experts in the subdomains of variational inference and sequential Monte Carlo methods.
- The authors should make clear how to explicitly implement the technique.
- The approach does not outperform state-of-the-art techniques in the Neural Latents Benchmark (NLB). While the authors mention that the "NLB metrics center around evaluating the quality of smooth rates inferred from spikes, which is not the central focus of our method. Rather, we aim to fit an RNN, from which -- by design -- we can sample noisy latent trajectories that reproduce variability in the data." But doesn't LFADS (and NDT?) allow generation of noisy latent trajectories (in the LFADS "factors") that reproduce variability in the data?

**Questions:**

- What is meant exactly by "tractable dynamics"? Is there a distinction between "tractable" and "interpretable"?
- Why is the proposed technique compared to Generalised Teacher Forcing in the EEG experiments, but not in the synthetic setups (Fig 3) or the spiking data (Figs 5-7)?

**Limitations:**

- The authors note just one assumption / limitation of their approach: an assumption of correlated Gaussian noise in the recurrent dynamics. How does the technique fair when observations from the true underlying system reflect private noise processes (e.g., measurement noise, variability in neurotransmitter release, etc)?
- One limitation that was not discussed or addressed is that the approach does not model the effects of unobserved inputs (beyond noting that correlated noise in the dynamics may arise due to unobserved inputs). Does this imply that the technique as presented is only appropriate for settings where the true dynamics can be reasonably modeled as an autonomous system?

---

> ### Author Rebuttal · Authors · 2024-08-06
>
> We thank the reviewer for recognizing that our approach "appears original in its combination of existing ideas” and that we provided “thoughtful” evaluations and demonstrations of our method. We clarified some raised issues below
> >Section 2.2 could benefit from being made more accessible to readers who are not experts in the subdomains of variational inference and sequential Monte Carlo methods.
> >The authors should make clear how to explicitly implement the technique.
>
> We would be happy to make these points more clear in the camera-ready version! We did provide additional implementation details in the appendix plus some of the code (which we by now also cleaned and re-organised, and will link in the camera-ready version).
>
> >But doesn't LFADS (and NDT?) allow generation of noisy latent trajectories (in the LFADS "factors") that reproduce variability in the data?
>
> The reviewer is correct that LFADS can also be used to generate latent trajectories (instead of inferring them), though it is not commonly used in this way. We now explored sampling from LFADS, both using the autonomous version, and when using the controller, after training on the rat HPC 2 dataset (Rebuttal Fig. C, D). In the case of an autonomous LFADS model, one samples an initial condition from the prior and then simulates a deterministic RNN forward. As on long sequences not all variability can be explained by variability in the initial condition, the latents end up not representing any variability that resembles the underlying system (Rebuttal Fig. C, cf. Manuscript Fig. 3). In the case of a full LFADS model with the controller, one can sample both an initial condition and time-varying inputs from the controller’s autoregressive prior. The full model seemed to rely overly on the controller’s data-inferred inputs, which deviated quite strongly from the samples from the controller’s autoregressive prior (Rebuttal Fig. D left). As a consequence, the latents do not seem to represent variability that is meaningful (Rebuttal Fig. D right).
>
> In general, accurately inferring firing rates from data does not necessarily translate to generating realistic data. NLB metrics do not evaluate generative quality, and in fact many high-performing methods in NLB are not used as generative models at all (e.g., NDT). We stress that performance on NLB is not the central focus of our method, and when it comes to generation our method outperformed relevant baselines.
>
> >What is meant exactly by "tractable dynamics"? Is there a distinction between "tractable" and "interpretable"?”
>
> With “tractable” we mean that properties of the dynamics, such as fixed points, can be obtained (exactly) in a computationally feasible way. The ability to compute fixed points of the dynamics efficiently can aid in interpretation and understanding the model’s underlying computations (see e.g., [1]). We will clarify this in the camera-ready version.
>
> >Why is the proposed technique compared to Generalised Teacher Forcing in the EEG experiments, but not in the synthetic setups (Fig 3) or the spiking data (Figs 5-7)?
>
> We thank the reviewer for pointing this out. We have now run GTF on one of the teacher student setups of Fig. 3A (see Rebuttal Fig. B). The deterministic student model that is fit with GTF doesn’t capture the latent dynamics of the stochastic teacher model well, and generated samples from the deterministic student are not as similar to the teacher model as when the student is fit with our method.
> As noted in the summary response, we did not compare GTF on data with Poisson observations, as GTF requires an invertible observation model, a limitation our method also overcomes.
>
> >How does the technique fair when observations from the true underlying system reflect private noise processes (e.g., measurement noise, variability in neurotransmitter release, etc)?
>
> Measurement noise that is unique to every neuron can be (and is in our manuscript) straightforwardly taken into account by choosing the appropriate observation model. Private noise that is iid over neurons and feeds back into the recurrency, reflecting e.g., variability in the neurotransmitter release, is however currently not modeled. There are cases where one can show that iid noise has little influence on the underlying population dynamics – e.g., in linear models with many more units than latent dimensions, where such noise becomes effectively orthogonal to the recurrent weights. However, as noted in our discussion, an interesting future direction would be investigating how to best incorporate private noise processes that feed into the recurrence.
>
> >Does this imply that the technique as presented is only appropriate for settings where the true dynamics can be reasonably modeled as an autonomous system?
>
> The reviewer raises a valid point. Our method does not explicitly model unobserved inputs. However, in principle, the model’s recurrent dynamics can capture unobserved inputs, as long as these inputs can be modeled as being generated by Gaussian noise, or by a subnetwork of the RNN (or in other words, as terms of a stochastic differential equation). Previous work on low-rank RNNs has focused on finding subpopulations in trained low-rank RNNs [2], which could potentially be applied here to find out which subpopulations in the trained RNN provide input to each other. Such an approach has been successfully applied to (deterministic) RNNs fit to trial-averaged data in [3].
>
> [1] Sussillo & Barak, 2013, Opening the black box: low-dimensional dynamics in high-dimensional recurrent neural networks
>
> [2] Dubreuil, Valente, Beiran, Mastrogiuseppe & Ostojic. 2022. The role of population structure in computations through neural dynamics
>
> [3] Valente, Pillow & Ostojic 2022. Extracting computational mechanisms from neural data using low-rank RNNs

---

> > ### Comment · Reviewer_Strk · 2024-08-12
> >
> > I appreciate the authors' thoughtful responses. I stand by my rating.

---

### Official Review · Reviewer_bWek · 2024-07-11

**Soundness:** 4
**Presentation:** 3
**Contribution:** 3
**Rating:** 6
**Confidence:** 4

**Summary:**

This paper proposes a low-dimensional, nonlinear dynamics model of neural data based on low-rank RNNs. In particular, the model dynamics are a discretization of the dynamics of a low-rank RNN in the space spanned by the column factors of the low-rank RNN matrix. The model is fit using variational SMC. The approach is validated in simulation and applied to three neural datasets of EEG, hippocampal, and motor cortex recordings. Additionally, for low-rank RNNs the authors introduce an exact fixed point finding procedure that runs in polynomial time.

**Strengths:**

The modeling approach and fitting methods are generally clearly described. The model is applied in multiple synthetic and real applications, where it recovers known ground truth or produces sensible results. Additionally, the proposed analytical technique for finding fixed points of low-rank RNNs appears to be generally useful for anyone working with such models.

**Weaknesses:**

It appears that the formulation of the model in the low-dimensional space loses some generality relative to the original low-rank RNNs. As the authors point out, for non-constant inputs the activity can lie outside the low-d subspace. While the proposed approximation in appendix C.2 may work for the settings in this paper, it is not necessarily clear how the approach would perform if the inputs to the original low-rank RNN are significantly non stationary.

All of the models fit in this paper have small latent dimensionalities (typically 2-3 latent dimensions). It is not clear how well the approach scales to larger dimensionalities than those considered here.

**Questions:**

- The authors point out in the discussion that the proposed model can be viewed as a neuralODE, when discussing the FINDR method. This paper could be improved by more clearly and directly discussing how the model and inference approach relate to other models and fitting methods in the neuralODE literature (another neuro example is ODIN from Versteeg et al). Since this model can be viewed as a neuralSDE, should it be using techniques from the neuralSDE literature for fitting?

- Generally, low-rank RNNs can have transient dynamics outside the column space of the weight matrix factors, e.g. via inputs as the paper mentions. The current approach may be limited to handling only certain types of inputs. I think it would be helpful for the authors to describe this difference and the limitations more clearly in the main text.

**Limitations:**

Yes

---

> ### Author Rebuttal · Authors · 2024-08-06
>
> We thank the reviewer for acknowledging that the paper is generally clear, and we hope to address their concerns, in particular with respect to when the inputs are time-varying.
> >It appears that the formulation of the model in the low-dimensional space loses some generality relative to the original low-rank RNNs. As the authors point out, for non-constant inputs the activity can lie outside the low-d subspace. While the proposed approximation in appendix C.2 may work for the settings in this paper, it is not necessarily clear how the approach would perform if the inputs to the original low-rank RNN are significantly non stationary
>
> We would like to clarify that our method also works in settings where the inputs are non-stationary, although unfortunately we did not describe or use this in the main paper. The reviewer is correct in that with time-varying inputs, the activity of the neurons in the network can lie outside of the column space of the recurrent weights: it will however still be constrained to be in the the space spanned by the columns of the recurrent and the columns of input weights (as we also describe in  first paragraph of supplement C.2). While for the models used in the paper, we indeed make an approximation ($\tilde{\mathbf{s}} \approx \mathbf{s}$), that allowed us to ignore the additional input dimensions, one does not need to make this approximation, and can simply consider the augmented system of $[\mathbf{z}, \tilde{\mathbf{s}}]$.
>
> We will explicitly include the equations for the distribution generated by the augmented system in the camera-ready version of the manuscript to clarify this.
>
> To explicitly demonstrate this, we now ran new experiments with a student-teacher setup, where a rank 1 teacher network was trained to report the sign of the mean of a time-varying noisy input (Rebuttal Fig. E).  We visualize the closely matching dynamics of the student and teacher network dynamics, in the plane spanned by the one column of the left recurrent connectivity vector $\mathbf{M}$ and the input weights vector (coordinates $\mathbf{z}$ and $\tilde{\mathbf{s}}$ respectively). This demonstrates that our method naturally works, even when there are time-varying inputs.
>
> >All of the models fit in this paper have small latent dimensionalities (typically 2-3 latent dimensions). It is not clear how well the approach scales to larger dimensionalities than those considered here.
>
> We would like to point out that our appendix already contained a model with 36 latents (Supplementary table 2). Additionally Rebuttal Fig. F contains full-rank models (rank 30 and rank 128) trained on the EEG dataset. We have since also trained new models on the whole rat HPC 11 dataset (instead of only the subset of data where the rat is moving), and we get good performance with 12-16 latents, we would be happy to include these in the camera-ready version of the paper.
>
> >The authors point out in the discussion that the proposed model can be viewed as a neuralODE, when discussing the FINDR method. This paper could be improved by more clearly and directly discussing how the model and inference approach relate to other models and fitting methods in the neuralODE literature (another neuro example is ODIN from Versteeg et al). Since this model can be viewed as a neuralSDE, should it be using techniques from the neuralSDE literature for fitting?
>
> We thank the reviewer for this comment, and will include a more extended discussion of related works as ODIN and neuralSDEs in our camera-ready version. A prominent line of work in the neural ODE/SDE literature concerns overcoming memory requirements of backpropagating through operations in an ODE solver, by for example using the adjoint method (i.e., substituting 'optimize then discretise' for 'discretise then optimize'). We here — similar to FINDR and ODIN — do not use the adjoint method, but rather a simple Euler-Maruyama discretisation scheme and 'standard' backpropagation through time. We note that this generally performed well for our use-cases. However, one could investigate how we can integrate our approach with variational approaches that use adjoint methods when fitting latent neural SDEs [1,2] as well as with filtering approaches for continuous time systems [3]. This could be especially relevant for irregularly sampled time-series.
> >Generally, low-rank RNNs can have transient dynamics outside the column space of the weight matrix factors, e.g. via inputs as the paper mentions. The current approach may be limited to handling only certain types of inputs. I think it would be helpful for the authors to describe this difference and the limitations more clearly in the main text.
>
> Please see our response to the first comment and Rebuttal Fig. E — our method naturally works with any kinds of inputs (irrespective of whether they are time-varying or not).
>
> References:
>
> [1]  Li, Wong, Chen & Duvenaud. 2020. Scalable Gradients for Stochastic Differential Equations
>
> [2] Deng, Brubaker, Mor & Lehrmann. 2021. Continuous Latent Process Flows
>
> [3] Särkkä & Sottinen. 2008. Application of Girsanov Theorem to Particle Filtering of Discretely Observed Continuous-Time Non-Linear Systems

---

> > ### Comment · Reviewer_bWek · 2024-08-12
> >
> > Thank you for your thorough response and additional experiments. It is nice to see that the method works well on a setting with time-varying inputs. Additionally, thank you for pointing out the experiments with larger latent dimensions. I am raising my score and recommending acceptance of this paper.

---

### Official Review · Reviewer_drM9 · 2024-07-11

**Soundness:** 4
**Presentation:** 4
**Contribution:** 4
**Rating:** 8
**Confidence:** 5

**Summary:**

The authors describe an elegant method to infer a low-dimensional description of stochastic neural dynamics using variational Sequential Monte Carlo and Low Rank RNNs. They apply this method to simulated data and to three experimental datasets (EEG, hippocampus, and motor cortex). They also describe an elegant analytic approach to determine the fixed points of the learned dynamical system.

**Strengths:**

The paper is very well-written. The motivating problem is well-established as an important question in computational neuroscience. The discussion and summary of prior work on this topic is executed very well. In addition to showing strong empirical results in simulation and real experimental analysis, the authors provide an illuminating mathematical analysis of low-rank RNNs and a procedure to efficient numerical procedure to find fixed points.

A key modeling advance is the ability to fit a model with stochastic and nonlinear dynamics. This is in contrast to most RNNs, which currently use a deterministic transition.

**Weaknesses:**

I enjoyed this paper and found very few weaknesses. However, I would ideally like to see the variational SMC method benchmarked against simpler baselines -- mostly I would be interested in SMC methods with simpler proposals (e.g. bootstrap particle filtering). Additionally, I would like to see more details about performance as a function of (a) number of particles, and (b) amount of observed data.

**Questions:**

n/a

**Limitations:**

The limitations are well discussed.

---

> ### Author Rebuttal · Authors · 2024-08-06
>
> We thank the reviewer for their positive feedback on our manuscript and their appreciation for modeling RNNs with stochastic dynamics! We have provided some additional analyses based on your suggestions, which we believe strengthens the original manuscript.
>
> >mostly I would be interested in SMC method with simpler proposals (e.g. bootstrap particle filtering).
>
> We included new experiments using the bootstrap proposal. When rerunning the student-teacher setup of Fig 3A, the student networks are not reliably able to recover the true latent noise of the teacher if the bootstrap proposal is used instead of the optimal proposal, and generally has a large variance in performance (Rebuttal Fig. G). When fitting models to the HPC-2 data, both the quality of the generated data, as well as the Hellinger distance between the power spectra of the latents with that of the local field potential is worse when the bootstrap proposal is used (Rebuttal Fig. H).
>
> >I would like to see more details about performance of (a) number of particles and (b) amount of observed data
>
> (a) We included new experiments where we vary the number of particles. Concretely, for the teacher student setup of Fig 3A, we show that with 1 particle (i.e., no resampling) the true underlying noise of the teacher network is not recovered, unlike when using 64 particles (Rebuttal Fig. G). When using 10 particles we still get close (in line with previous work on variational sequential monte carlo, where it is suggested to use a small number of particles during training for efficiency, and then a larger number for evaluation). For the HPC-2 dataset, we similarly obtain more informative latents when using multiple particles (Rebuttal Fig. H).
>
> (b) We also included a new experiment where we ran both our method and GTF on 30 seconds (instead of 60) of the EEG data (Rebuttal Fig. I).  While both methods have a slightly worse KL-divergence score ($\mathsf{D\_{stsp}}$), this is in line with train/test shift between the first and second 30s of the EEG data. Additionally, we included a new student-teacher experiment (Rebuttal Fig. A) where the student only sees half of the teacher's neurons and still learns to approximate the teacher’s latent dynamics.

---

> > ### Comment · Reviewer_drM9 · 2024-08-09
> > **Reviewer Response**
> >
> > Thanks for the additional experiments. I retain my positive score.

---

### Official Review · Reviewer_nuem · 2024-07-13

**Soundness:** 3
**Presentation:** 3
**Contribution:** 2
**Rating:** 5
**Confidence:** 4

**Summary:**

This work focused on inferring stochastic low-rank structure from neural data. It developed a low-rank RNNs as state space models, and using sequential monte Carlo (SMC)  to learn the model's parameters. The proposed model is efficient in finding all fixed points in a polynomial cost instead of exponential cost. It is evaluated on multiple settings, one from student-teacher setups, given ground truth teacher RNN, and infer the structure and statistics, and demonstrated on extracting latents from EEG data, and recover interpretable latent dynamics from spike recordings in hippocampus, and allow position decoding.

**Strengths:**

**Motivation**
This work focused on important question in neuroscience, given partial observation and initial states, building stochastic models is essential to model complex neural dynamics.

**Method**
The proposed method is efficient in terms of the low-rank structure, and as well as low complexity in finding fixed points.

**Evaluation**
The method is demonstrated on extensive neural datasets, as well as synthetic networks. Evaluations are compared with deterministic model GTF.

**Weaknesses:**

**Novelty**
Low-rank RNN has been demonstrated on many applications in recent works, the novelty is limited.

**Experiments**
Adding simulation with partial observations to demonstrate the effectiveness of stochastic low-rank RNN.

**Results**
Limited performance in recovering latent structure and predicting dynamics,  how does GTF perform in this task?
1. Fig 3 a and b.
2. Fig 4, the generated trace is not similar to EEG signal ground truth in the qualitative evaluation.
3. Fig 5c, pairwise correlation is lower than 0.10.
4. Fig 7e, the mismatch between true and generated in mean ISI dissimilarity.

**Questions:**

1. Explain the limited performance of model as mentioned above, how does GTF perform in the same task?

2. Ablation studies of stochasticity and low-rank constraints.

**Limitations:**

No potential negative societal impact.

---

> ### Author Rebuttal · Authors · 2024-08-06
>
> We appreciate the reviewer's acknowledgement of the importance of the question we focus on, and the efficiency of our proposed methods for fitting low-rank models and finding their fixed points. We here hope to clarify some one of the raised concerns.
>
> >Low-rank RNN has been demonstrated on many applications in recent works, the novelty is limited.
>
> While there has indeed been extensive applications of deterministic RNNs, we here provide a method for fitting *stochastic non-linear* low-rank RNNs to single-trial neural data.  We also show the importance of using stochastic RNNs to capture observed neural variability.  Rebuttal Fig. B-D in the attached pdf highlight the importance of being able to infer the right level of latent noise. Related works (that we also cite) consider stochastic *linear* low-rank RNNs [1] or fit *deterministic* low-rank RNNs to trial-averaged data [2].
>
>
> >Adding simulation with partial observations to demonstrate the effectiveness of stochastic low-rank RNNs
>
> The low-rank setting naturally allows for partial observations. We have added a new experiment where we only observe 10 of the 20 units (Rebuttal Fig. A). The ground truth dynamics are still accurately captured.
> >Limited performance in recovering latent structure and predicting dynamics, how does GTF perform in the same task
> >Explain the limited performance of the model as mentioned above, how does GTF perform in the same task
>
> We disagree with the reviewer's assessment of  "limited performance". In our student teacher setups we accurately capture the true latent dynamics (as well as the true level of latent noise, see Rebuttal Fig. G), and our empirical results on e.g., EEG data match state-of-the art. As noted in the summary response, we did not compare GTF on data with Poisson observations as GTF requires an invertible observation model, a limitation our method also overcomes. We have, however, now run GTF (deterministic) on one of the teacher-student setups with continuous observations, where (unlike our method) it  fails to capture the stochastic dynamics of the teacher model (Rebuttal Fig. A).
>
> >Fig 5c, pairwise correlation is lower than 0.10.
>
> Fig 5c shows the pairwise correlation between neurons’ activity, plotted against the pairwise correlation between neurons of our fit network. The fact that spikes of real neurons have a low correlation with each other (the values of which our model captures) can hardly be seen as a limitation of our method. We will clarify this plot in the camera ready version.
> >Fig 4, the generated trace is not similar to EEG signal ground truth in the qualitative evaluation
>
> While we agree with the reviewer that there is no perfect match between generated and ground truth traces, we note that quantitatively the (smoothed) traces generated by our model match state-of-the art, while needing only 3 latent dimensions. The best-performing method in [3] used 16 latents.
> >Fig 7e, the mismatch between true and generated in mean ISI dissimilarity
>
> We note that the ISI distribution is generally relatively noisy, see for instance the shift between mean train and test ISI per condition (supplementary figure 6), which can be in the 100s of ms for some neurons. We would argue that, given this, we do capture the differences in distributions between conditions well: The median absolute deviation of off-diagonal elements in the dissimilarity matrices is below .1 between our simulated data and the test set.
> >ablation studies of low-rank constraints
>
> We have now fit full-rank RNNs to the EEG dataset, both models with 30 units (which have a similar number of parameters to our rank 3, 512 unit RNN), and models with 128 units (which have over 10x more parameters). We note that the KL divergence between data and simulated data is higher for the full rank models, while additionally being less tractable than our low-rank RNN (Rebuttal Fig. F).
> >ablation studies of stochasticity
>
> While our method is fundamentally probabilistic, GTF can be used to fit deterministic low-rank RNNs (we describe the relation in the manuscript section 2.2.2). We now show that when the true underlying model is stochastic, fitting a deterministic model with GTF does not allow one to accurately capture the true underlying model (Rebuttal Fig. B). Besides this, empirically, on the EEG dataset, stochasticity allowed us to obtain a reconstruction similar to GTF with only 3 latents, as opposed to having to learn a complex deterministic 16-dimensional chaotic system.
>
> [1] Valente, Pillow & Ostojic 2022. Probing the Relationship Between Latent Linear Dynamical Systems and Low-Rank Recurrent Neural Network Models
>
> [2] Valente, Pillow & Ostojic 2022. Extracting computational mechanisms from neural data using low-rank RNNs

---

> ### Comment · Reviewer_nuem · 2024-08-12
>
> Thanks for authors' efforts in adding ablation studies for stochasticity and low-rank constraints in Fig 1A, B F. My concerns related to these questions have been adequately addressed. While I still have reservations about the mismatch between the generated traces and GT, shown in Fig 4. I am convinced that it is comparable to the current SOTA, with more parameter-efficiency as an advantage, while I still expect further improvements with more advanced methodologies. I have increased my score correspondingly.

---

### Author Rebuttal · Authors · 2024-08-06

**General response**

We thank the reviewers for their extensive comments and insightful feedback on our manuscript. Our paper introduced a method for obtaining low-dimensional descriptions of stochastic neural dynamics, tackling a “well-established (drM9)” and “important question in neuroscience” (neum). Additionally, we proposed a method for finding fixed points in piecewise-linear low-rank RNNs, using an “elegant analytic approach” (drM9) that “appears to be generally useful for anyone working with such models” (bWek).

The reviewers  appreciated the "strong empirical results" (drM9) that “improve upon state-of-the-art techniques (e.g., GTF) in certain settings" (Strk), which were ”demonstrated on extensive neural datasets, as well as synthetic networks" (neum). Moreover the reviewers provided positive feedback on the presentation, stating that we provided an "illuminating mathematical analysis of low-rank RNNs" (drM9) and that the "figures and tables are clearly presented and quite interpretable" (Strk).

Nevertheless, the reviewers highlighted important points which we have now addressed with new figures and explanations. We here summarize the main new analyses, which fully support our original claims.


**Importance of fitting stochastic models**

Both reviewer neum and Strk ask for more comparisons to generalised teacher forcing  (GTF) [1] — a related method for fitting RNNs with deterministic transitions, as we discuss in the manuscript. We repeated one of the student-teacher setups by fitting a student with GTF, which demonstrates that — if the underlying RNN is indeed stochastic — deterministic GTF is not adequate (Rebuttal Fig. 1B). Thus, one needs methods for stochastic dynamics  (such as the one we propose), both to obtain good reconstructions and to recover the true latent dynamics. This point is also reinforced by a new experiment where we fit LFADS [2] to the HPC2 dataset (Strk; Rebuttal Fig. 1C,D).
Furthermore, in the formulation of GTF in [1], the authors require an invertible observation model (see section 3.4 in [1]). We can therefore not directly compare our method to GTF in the experiments with spiking observations (Manuscript Figs 5-7). Indeed, one advantage of our method is that it overcomes the limitation of needing an invertible observation model (by using an encoding network that predicts a distribution over latents).


**Additional analysis of hyper-parameters and training setup**

We performed additional experiments where we demonstrate that:
- In a teacher-student setup, we can recover the underlying latent dynamical system, even when we only fit to  partial observations (neum; Rebuttal Fig. 1A).
- We can recover the true latent noise in student teacher setups, when we use the optimal proposal ($ p(\mathbf{z}\_t | \mathbf{y}\_t, \mathbf{z}\_{t-1}) $ ) with 64 particles, and can get close when using 10 particles. When using a bootstrap proposal ($ p(\mathbf{z}\_t|\mathbf{z}\_{t-1})$) or only 1 particle (i.e., no resampling) this is no longer the case (drM9; rebuttal Fig 1G). The bootstrap proposal also leads to worse performance on the HPC2 dataset (drM9; rebuttal Fig 1H). This indicates that our strategy of using multiple particles and conditioning the proposal distribution on observed data, is indeed beneficial relative to alternatives.
- We fit full-rank models (32 and 128 units) to the EEG dataset, and show that the performance of full rank RNNs is worse than those of  low-rank RNNs, on top of the models being less tractable (neum; Rebuttal Fig. 1F).


**Generalization to time-varying inputs**

Reviewer bWek asks how the approach would perform if the inputs to the original low-rank RNN are non stationary. We unfortunately did not describe or use this in the main paper, but our method works with time-varying inputs (see the equations in the first paragraph of Supplement C.2 which define a low-rank RNN with arbitrary inputs). Our probabilistic setup therefore also allows fitting models with time-varying inputs. To demonstrate this, we now added a new teacher-student setup where the teacher network integrated a time-varying stimulus (Rebuttal Fig. 1E), and show that the student network successfully learned the latent dynamics of the teacher using our method. We will update the paper accordingly.

[1] Hess, Monfared, Brenner & Durstewitz. 2023. Generalized Teacher Forcing for Learning Chaotic Dynamics
[2] Pandarinath, O’Shea, Collins, Jozefowicz, Stavisky, Kao, Trautmann, Kaufman, Ryu, Hochberg, Henderson, Shenoy, Abbott & Sussillo. 2018. Inferring single-trial neural population dynamics using sequential auto-encoders

---

### Decision · Program_Chairs · 2024-09-25

**Decision:**

Accept (poster)

**Comment:**

This work introduces stochastic low-rank RNNs, trained by sequential Monte-Carlo in a way that is related to generalized teacher forcing. The authors further describe a nice technique for finding all fixed points of the low-rank model in polynomial rather than exponential time. In addition to explicitly accounting for noise in the process, the proposed method finds much lower-dimensional latent spaces which is a major plus. While some issues with novelty and lack of comparisons were brought up in the reviews, these were largely addressed in the rebuttal, and all referees finally supported acceptance. I fully agree and think this paper is not only technically strong and solid, but contributes substantial advances to the field.

Just a little note from my side: The authors claim (and show for a very particular example) that GTF cannot fit stochastic dynamics, but in the cited paper it is indeed trained on systems with noise and seems to perform quite well?